# The phytoplasma SAP54 effector acts as a molecular matchmaker for leafhopper vectors by targeting plant MADS-box factor SVP

**Zigmunds Orlovskis*[†], Archana Singh[‡], Adi Kliot[§], Weijie Huang, Saskia A Hogenhout***

John Innes Centre, Norwich Research Park, Norwich, United Kingdom

**\*For correspondence:**
zigmunds.orlovskis@biomed.lu.lv (ZO);
saskia.hogenhout@jic.ac.uk (SAH)

**Present address:** [†]Latvian Biomedical Research and Study Centre, Rātsupītes 1k-1, Rīga, Latvia; [‡]Department of Plant Sciences, University of Cambridge, Downing Street, Cambridge, United Kingdom; [§]Department of Entomology, ARO, the Volcani Center, HaMaccabim Road 68, 14 PO Box 15159, Rishon LeZion, Israel

**Competing interest:** The authors declare that no competing interests exist.

## eLife Assessment

This study highlights an **important** discovery: a bacterial pathogen's effector influences plant responses that in turn affect how the leafhopper insect vector for the bacteria is attracted to the plants in a sex-dependent manner. The research is backed by **convincing** physiological and transcriptome analyses. This study unveils a complex interdependence between the pathogen effector, male leafhoppers, and a plant transcription factor in modulating female attraction to the plant, shedding light on previously unexplored aspects of plant-bacteria-insect interactions.

**Abstract** Obligate parasites often trigger significant changes in their hosts to facilitate transmission to new hosts. The molecular mechanisms behind these extended phenotypes - where genetic information of one organism is manifested as traits in another - remain largely unclear. This study explores the role of the virulence protein SAP54, produced by parasitic phytoplasmas, in attracting leafhopper vectors. SAP54 is responsible for the induction of leaf-like flowers in phytoplasma-infected plants. However, we previously demonstrated that the insects were attracted to leaves and the leaf-like flowers were not required. Here, we made the surprising discovery that leaf exposure to leafhopper males is required for the attraction phenotype, suggesting a leaf response that distinguishes leafhopper sex in the presence of SAP54. In contrast, this phytoplasma effector alongside leafhopper females discourages further female colonization. We demonstrate that SAP54 effectively suppresses biotic stress response pathways in leaves exposed to the males. Critically, the host plant MADS-box transcription factor short vegetative phase (SVP) emerges as a key element in the female leafhopper preference for plants exposed to males, with SAP54 promoting the degradation of SVP. This preference extends to female colonization of male-exposed *svp null* mutant plants over those not exposed to males. Our research underscores the dual role of the phytoplasma effector SAP54 in host development alteration and vector attraction - integral to the phytoplasma life cycle. Importantly, we clarify how SAP54, by targeting SVP, heightens leaf vulnerability to leafhopper males, thus facilitating female attraction and subsequent plant colonization by the insects. SAP54 essentially acts as a molecular 'matchmaker', helping male leafhoppers more easily locate mates by degrading SVP-containing complexes in leaves. This study not only provides insights into the long reach of single parasite genes in extended phenotypes, but also opens avenues for understanding how transcription factors that regulate plant developmental processes intersect with and influence plant-insect interactions.

**eLife digest** The parasitic bacterium phytoplasma is a master manipulator. It turns its hosts into sterile 'zombie' plants that remain alive only to support the parasite. Phytoplasma secretes a protein called SAP54, which transforms the flowers of the plants into leaf-like structures. Until recently, scientists believed this transformation was how the parasite attracted tiny sap-feeding insects called leafhoppers, which act as vectors that transport phytoplasma to its next host. However, more recent research has shown that the transformation of flowers into leaf-like structures is not needed to lure leafhoppers to the plant. So, what is it about SAP54 that manipulates the preferences of leafhoppers?

To find out, Orlovskis et al. genetically modified plants to produce SAP54. They then carefully observed the number of male and female leafhoppers attracted to the mutant plants compared to plants that had not been genetically manipulated. They also used genetic analysis to investigate the proteins controlling the plant's defence mechanisms.

Orlovskis et al. found that SAP54 attracted female leafhoppers to the leaves of the plant, but only when males were also present. SAP54 also suppressed the plant's defences when males were on the leaves, making the plant more inviting to females. Increasing the number of females naturally facilitates breeding, resulting in more insects that can transport the parasite to new host plants.

A plant protein, called SHORT VEGETATIVE PHASE (or SVP for short), turned out to be critical in this process. Orlovskis et al. discovered that SAP54 promotes the breakdown of SVP, and plants lacking this protein also attracted more females when exposed to male leafhoppers. This suggests that SAP54 acts as a 'molecular matchmaker', helping male leafhoppers find mates by breaking down SVP in leaves. The involvement of SVP in attracting leafhoppers is surprising, as the family of proteins which SVP belongs to is primarily known for regulating developmental processes, such as flowering, rather than influencing how plants interact with their environment.

These findings deepen our understanding of the complex relationships between parasites, plants and insects, demonstrating how parasites manipulate both plant biology and insect behaviour. This knowledge could inform new strategies for controlling plant diseases spread by insects, potentially reducing crop losses caused by phytoplasma.

## Introduction

Parasites often excel in altering the development and behavior of their hosts, a trait essential for their survival and propagation. The trait is particularly pronounced in obligate parasites, which have substantial control over their hosts, earning them the nickname 'puppet masters'. The phenomenon exemplifies the concept of extended phenotypes (originally coined by *Dawkins, 1982*), where the impact of an organism's genes extends beyond its own physical form and affects other organisms. This concept is especially evident in obligate parasites that depend on alternate hosts for transmission, as they often do not only alter the conditions of their immediate hosts but also have far-reaching effects on other organisms in the ecosystem.

For instance, the protozoan parasite *Taxoplasma gondii* changes the behavior of their intermediate rodent hosts by reducing their innate fear for cats, which are the definite hosts where the parasite undergoes sexual reproduction to produce oocysts (*Tong et al., 2021*). Similarly, the rust fungus *Puccinia monoica* induces its host plant, *Boechera stricta*, to produce 'pseudoflowers'. These structures mimic real flowers and attract pollinators with their scent and sugary rewards, an essential strategy for the fungus to spread its spores between plants (*Roy, 1993*; *Cano et al., 2013*). Although advancements have been made in identifying parasite virulence factors, our understanding of the specific host processes that are commandeered to produce these extended phenotypes is still developing.

Parallel to these discoveries are the studies on the extended phenotypes of phytoplasmas, which have revealed that their manipulative abilities stem from particular genes within these pathogens (*Sugio et al., 2011b*; *Tomkins et al., 2018*; *Huang et al., 2020*; *Wang et al., 2024*). Phytoplasmas cause disease in crops, ornamentals and native plants worldwide (*Kumari et al., 2019*) and are often dependent on sap-feeding insects, including leafhoppers, plant hoppers and psyllids, for transmission (*Weintraub and Beanland, 2006*). As obligate bacterial parasites, phytoplasmas frequently trigger the emergence of unusual plant structures such as leaf-like floral parts (phyllody) and the excessive growth and clustering of leaves and branches (witch's broom; *Lee et al., 2000*; *Al-Subhi et al., 2018*;

*Kumari et al., 2019*). These alterations not only compromise plant health but also promote attraction and colonization of insect vectors that are primarily responsible for phytoplasma spread and transmission (*Sugio et al., 2011a*; *Frost et al., 2013*; *MacLean et al., 2014*; *Orlovskis and Hogenhout, 2016*; *Clements et al., 2021*; *Al-Subhi et al., 2021*; *Huang and Hogenhout, 2022*). The responsible phytoplasma genes encode for effector molecules that, once inside the plant cell cytoplasm, target and typically disrupt or degrade essential plant transcription factors involved in growth, development, and defence (review by *Wang et al., 2024* and references therein as well as *Liu et al., 2023*; *Suzuki et al., 2024*; *Correa Marrero et al., 2024*; *Yan et al., 2024*; *Zhang et al., 2024*). This molecular interference exemplifies the extended phenotype reach, affecting not just the host appearance but also its physiological and biological processes.

The interaction between the Aster Yellows strain Witches Broom (AY-WB) phytoplasma and its vector, the aster leafhopper *Macrosteles quadrilineatus*, offers insights into the complex interplay between parasites and hosts. The effector protein secreted AY-WB protein (SAP) 11 binds to and destabilizes class II TCP transcription factors - this action leads to changes in leaf shapes and stem proliferation, reminiscent of witches' brooms, and altered root architecture resembling the hairy roots found in infected plants (*Bai et al., 2009*; *Sugio et al., 2011a*; *Sugio et al., 2014*; *Lu et al., 2014*; *Chang et al., 2018*; *Pecher et al., 2019*). Additionally, this effector reduces plant jasmonic acid and salicylic-acid-mediated defence responses, alters volatile organic compounds, and promotes the reproduction rates of the insect vectors (*Sugio et al., 2011a*; *Lu et al., 2014*; *Tan et al., 2016*). Another effector, SAP05, recruits the 26 S proteasome component RPN10, instigating the breakdown of SPL and GATA transcription factors (*Huang et al., 2021*; *Liu et al., 2023*; *Yan et al., 2024*; *Zhang et al., 2024*). This process is linked to the typical leaf and stem proliferations of witches' brooms in phytoplasma-infected plants. The SAP05-mediated degradation of SPLs, but not GATAs, increases the attractiveness of these plants to insect vectors (*Huang et al., 2021*).

However, the relationship between phytoplasma SAP54/PHYL1 effectors inducing leaf-like flowers and attracting insects seems more intricate. These effectors target and disrupt MADS-box transcription factors, akin to animal HOX genes, leading to a transformation of flowers into leaf-like structures, alterations that mirror the phyllody and virescence symptoms of phytoplasma infection (*MacLean et al., 2011*; *MacLean et al., 2014*; *Wang et al., 2024*; *Suzuki et al., 2024*; *Correa Marrero et al., 2024*). SAP54 recruits plant 26 S proteasome shuttle factors called radiation sensitive 23 (RAD23) to break down these transcription factors (*MacLean et al., 2014*; *Kitazawa et al., 2022*; *Suzuki et al., 2024*). Leafhoppers prefer plants that stably produce SAP54 for reproduction and this preference phenotype requires the presence of RAD23, which is essential for both breaking down MADS-box transcription factors and inducing leaf-like flowers (*MacLean et al., 2014*). However, the increased reproduction of insect vectors does not depend on the presence of leaf-like flowers. Our past research showed that insects prefer SAP54 plants during the vegetative developmental phase, before flowers emerge, and even flowering SAP54 plants upon removal of the leaf-like flowers (*Orlovskis and Hogenhout, 2016*). Therefore, it remains unclear whether plant MADS-box transcription factors play a role in the leafhopper attraction phenotype.

MADS-box transcription factors have been predominantly studied in relation to regulation of floral transition and flower organ development. Nonetheless, factors like suppressor of overexpression of CO 1 (SOC1) and short vegetative phase (SVP) are also expressed in leaves and other organs during the *A. thaliana* vegetative developmental phase, and regulate processes other than orchestrating the flowering process, such as plant defence responses, as has been shown for SOC1 (*Li et al., 2020*). Based on these insights, we hypothesize that leafhopper attraction is influenced by MADS-box transcription factors involved in processes other than flowering.

Here, we analysed the factors influencing leafhopper colonization preference in SAP54 plants. Surprisingly, we found that colonization preference on SAP54 plants only occurs in the presence of males on leaves during choice tests. Both SAP54 and males are essential for female preference, as females did not exhibit a preference for male-exposed versus non-exposed control plants, nor for SAP54 plants versus control plants in the absence of males. In contrast, female-only presence on SAP54 plants deters colonization by other females. We noted a clear downregulation of biotic stress response pathways in male-exposed SAP54 leaves, contrasting with the upregulation observed in female-exposed SAP54 leaves and control plants exposed to both males and females. Furthermore, we found that the MADS-box transcription factor SVP is essential for female preference in male-exposed

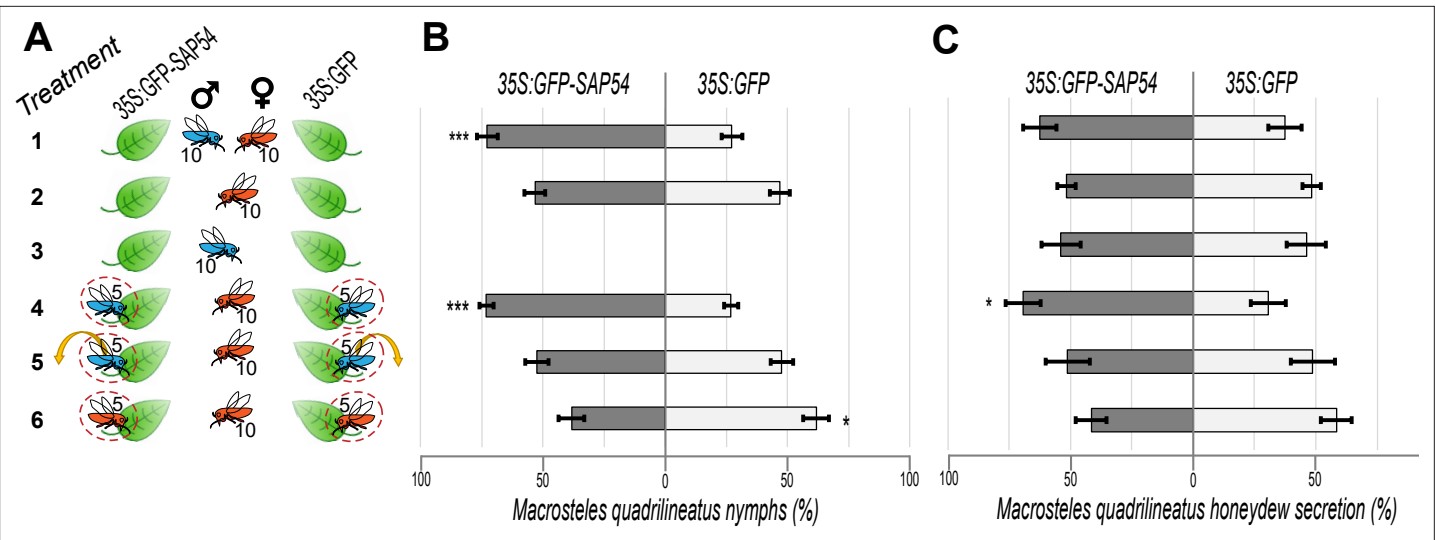

**Figure 1.** *M. quadrilineatus* leafhopper preference to reproduce and feed on SAP54 versus GFP plants is dependent on leaf exposure to leafhopper males. (**A**) Experimental design of 6 choice tests (treatments) with 10 male and/or 10 female insects, as indicated, on 6 weeks old *A. thaliana* rosettes. Dashed circles indicate clip-cages and the arrow indicates the removal of males before the start of the choice test. Each choice test (treatment) is placed in a separate cage. (**B**) Percentages (%) of nymphs found on SAP54 versus GFP plants. (**C**) Percentages (%) of leafhopper honeydew secretions in choice tests where insects are allowed to feed on either SAP54 or GFP plants. Horizontal bars in (**B, C**) indicate the mean ± 1 SEM. *p<0.05, ***p<0.001. The entire series of choice tests 1–6 were performed in parallel and repeated independently three times for progeny count and two times for honeydew quantification - data presented in (**B**) and (**C**) include the pooled results of the independent choice test series.

The online version of this article includes the following figure supplement(s) for figure 1:

**Figure supplement 1.** Data distributions of independent repeats that were used to generate graphs displayed in *Figure 1B and C*.

**Figure supplement 2.** *Macrosteles quadrilineatus* female leafhoppers show no preference for *A. thaliana* Col-0 wild-type plants exposed to conspecific male leafhoppers.

**Figure supplement 3.** Female *M. quadrilineatus* preference for male-exposed SAP54 plants are unlikely to involve long-distance cues.

SAP54 plants. Females also prefer colonizing male-exposed *svp null* mutant plants over non-exposed ones. Our research sheds light on how the conserved MADS-box transcription factor SVP influences leaf susceptibility to male herbivorous insects, promoting female attraction and colonization.

## Results

### Attraction of SAP54 plants to fecund female leafhoppers relies on the simultaneous presence of leafhopper males on these plants

Given our previous finding that SAP54 promotes fecundity of the AY-WB phytoplasma leafhopper vector *M. quadrilineatus*, we wished to further investigate what aspect of the insect-plant interaction is affected by SAP54. Females feed and lay eggs, while males only feed, and it is known that plants can induce different plant defence responses to insect feeding and egg laying (*Little et al., 2007*). Hence, the first experiment was to assess the SAP54 effect on males and females separately using choice tests. In these tests, the insects were given a choice to feed and lay eggs on 35 S:SAP54 or 35 S:GFP transgenic plants (henceforth referred to as SAP54 and GFP plants) for 5 days, then adult leafhoppers were removed. Immediately after, plants from each choice test were individually caged to avoid the hatching nymphs from moving between plants. The number of progeny (nymphs) were counted 14 days later. When these choice tests were done with both males and females, the leafhoppers produced more progeny on SAP54 than on GFP plants (treatment 1 in *Figure 1A–B* and *Figure 1—figure supplement 1*), confirming previous findings (*MacLean et al., 2014*; *Orlovskis and Hogenhout, 2016*). However, to our surprise, choice tests with fecund females alone, without males, did not result in more leafhopper progeny on the SAP54 versus GFP plants (treatment 2 in *Figure 1A–B* and *Figure 1—figure supplement 1*). The females also produced more progeny on the SAP54 compared to GFP plants when the males were caged and were not in direct contact with

females (treatment 4 in *Figure 1A–B* and *Figure 1—figure supplement 1*). In contrast, the females did not produce more progeny on the SAP54 plants when males were left on the plant for only 48 hr and removed before the choice test with females alone (treatment 5 in *Figure 1A–B* and *Figure 1—figure supplement 1*) or when the plants were exposed to the conspecific females in the clip cages during fecund female-only choice tests (treatment 6 in *Figure 1A–B* and *Figure 1—figure supplement 1*). In the latter choice test, there was a reproduction preference for the GFP plants instead. These results indicate that the presence of males on the leaves are required for the leafhopper preference to reproduce more on SAP54 versus GFP plants. Importantly, we found no obvious difference in progeny when fecund females were given a choice between (non-transgenic) wild type plants with and without males (*Figure 1—figure supplement 2*), indicating that SAP54 is required for the modulation of plant processes that results in the male-dependent reproduction preference of females.

Quantification of honeydew excretions provide a measure of how much phloem fluids the insects acquire from their host plant and therefore is a proxy for measuring feeding preferences (*Hong and Rumei, 1993*; *Ammar et al., 2013*; *Cameron et al., 2014*). Honeydew secretion tests showed that fecund females secreted more honeydew on SAP54 plants in the presence of males in clip-cages (treatment 4 in *Figure 1C* and *Figure 1—figure supplement 1*), whereas no increase in honeydew secretions of these females was observed on the SAP54 plants in absence of males (treatment 2 in *Figure 1C* and *Figure 1—figure supplement 1*) or when males were removed from the clip-cages prior to the choice test with females alone (treatment 5 in *Figure 1C* and *Figure 1—figure supplement 1*), indicating that females ingest a greater volume of plant sap from SAP54 than from GFP plants when males are present. There were slight increases in honeydew secretions on SAP54 versus GFP plants when the choice tests were done with both sexes, but these differences were not significant (treatment 1 in *Figure 1C* and *Figure 1—figure supplement 1*), in agreement with behavior studies showing that female feeding is interrupted when in direct contact with males possibly due to mating (*Beanland et al., 1999*). Females did not produce more honeydew on the SAP54 plants when other conspecific females were caged on the same plant (treatment 6 in *Figure 1C* and *Figure 1—figure supplement 1*), indicating their enhanced feeding activity is specifically dependent on the presence of males. Males alone did not secrete more honeydew on SAP54 versus GFP plants (treatment 3 in *Figure 1C* and *Figure 1—figure supplement 1*), indicating that males alone do not show any feeding preference for SAP54 plants and that the females are primarily responsible for the increased honeydew production on male colonized SAP54 plants.

Together, the fecundity and honeydew secretion data indicate that attraction of SAP54 plants to fecund female leafhoppers relies on the simultaneous presence of leafhopper males on the plants. The females produce more progeny and feed more on these plants only when males are present, even when the males are physically separated from the females.

## Female leafhopper preference for male-exposed SAP54 plants unlikely involves long-distance cues

To investigate if the fecund females are attracted to the SAP54 plants via volatile cues released by male-exposed SAP54 plants or mating calls from males that may be perceived by females from a distance, we established a sticky-trap assay that capture females before they access the plants themselves. Interestingly, female leafhoppers were equally likely captured on GFP and SAP54 plants when both of these plants were placed in an odour and sound permeable black container, equipped with either transparent or green sticky traps (*Figure 1—figure supplement 3A* choice tests 1 and 2), suggesting that volatile or acoustic cues emitted from male-exposed GFP or SAP54 plants do not bias female landing choice at a distance when identical visual cues are present. To test whether the assay works, we compared different color sticky traps as a positive control for insect visual choice. By doing so, female leafhoppers were more likely to be captured on green versus transparent sticky traps placed over the black containers, regardless of whether the green sticky traps were placed over male exposed GFP or SAP54 plants (*Figure 1—figure supplement 3A* choice tests 3 and 4), indicating that visual cues from the green traps were the primary determinant for female landing choice. Female insects also did not show an obvious landing preference for wild-type plants colonized by males compared to insect-free plants (*Figure 1—figure supplement 3B*). These results indicate that female choice for SAP54 versus GFP plants does not appear to depend solely on plant volatiles that are released by the SAP54 plants in the absence or presence of males on the leaves and that the

females do not appear to respond to potential mating calls of the males that may be perceived by the females from a distance. Therefore, these results indicate that female reproductive preference for the male-exposed SAP54 versus GFP plants is dependent on the direct females access to the leaves of SAP54 plants and presence of males on these leaves.

## SAP54 plants display a dramatic altered transcriptional response specifically to male leafhoppers

To further investigate how the combination of SAP54 and male-exposure may affect the preference of fecund females to reproduce on SAP54 versus GFP plants, we conducted RNA-seq experiments and compared the transcriptional responses of leaves of SAP54 and GFP plants that were exposed to five individual *M. quadrilineatus* females or males in clip cages or to clip cages without insects (controls). Each treatment consisted of four biological replicates (*Figure 2—figure supplement 1A*). After normalization, transcripts with coverage of <1 FPKM in all replicates in at least one of the treatments were removed, leaving 16,307 genes (*Figure 2—figure supplement 1B*). In a separate analysis, the data were analysed for differentially expressed genes (DEGs) that display significant changes in transcription levels between any two treatments, amounting to 6957 DEGs (*Figure 2—figure supplement 1B*). Considering both categories, a total of 17,153 genes were identified (*Supplementary file 1*) of which 6111 genes were shared between the categories (*Figure 2—figure supplement 1B*). Based on these 17,153 genes, we performed multiple discriminant analysis (MDA) to visualize the separation between the treatment groups. While most treatments largely clustered together, SAP54_MALE_2 and GFP_MALE_2 appear to be switched based on the groupings of the other three replicates in the SAP54_male and GFP_male treatments (*Figure 2—figure supplement 1C*). Analyses of the normalized reads that map to GFP versus GFP-SAP54 ruled out that the plant genotypes were accidentally swapped between the SAP54_MALE_2 and GFP_MALE_2 treatments or that the GFP or GFP-SAP54 were not expressed in the leaves (*Figure 2—figure supplement 1D*). Moreover, plots of the median FRKM of all transcripts versus the GFP or GFP-SAP54 read counts did not flag the SAP54_MALE_2 and GFP_MALE_2 treatments as different from the others (*Figure 2—figure supplement 1E*). Therefore, the apparent SAP54_MALE_2 and GFP_MALE_2 outliers may represent inherent variation of the biological system. Nonetheless, the MDA plot shows that in at least three out of four replicates per treatment, the leaf transcriptional responses are clearly distinct in male-treated SAP54 plants versus GFP plants, and more so than for the non-exposed and female-exposed SAP54 plants versus GFP plants (*Figure 2—figure supplement 1C*).

When all samples were considered, a total of 136 DEGs were identified (*Figure 2—figure supplement 2A*). Of these, 79 DEGs were derived from male-exposed leaves of SAP54 plants versus those of GFP plants and 69 DEGs from the non-exposed leaves of SAP54 versus those of GFP plants, with 12 DEGs overlapping between these two categories, whereas no DEGs were identified from female-exposed leaves of SAP54 versus those of GFP plants (*Figure 2—figure supplement 2A*). This suggests that SAP54 has a bigger impact on leaf responses to males than to females. This was corroborated with the PCA analysis showing that insect exposure, male or female, of leaves of either SAP54 or GFP plants explained 54% of the variance (PC1), whereas the male-exposed SAP54 leaves are predominantly responsible for the 25% variance of PC2 among the insect-exposed treatments (*Figure 2—figure supplement 2B*). The impact of SAP54 on male-exposed leaves was more obvious when three outliers were removed (*Figure 2—figure supplement 2C and D*). Because the goal of this experiment was to assess how leaf responses of male-exposed plants differ from other treatments, we excluded these outliers, as including these obscures the transcriptional signal derived from the other three samples.

To further assess how SAP54 modulates leaf responses specifically to males, we compared DEGs between male or female exposed leaves of GFP versus those of SAP54 plants (*Figure 2A*; *Supplementary file 2*). The leaf response of GFP plants to female exposure amounted to 2375 DEGs as opposed to 909 male-responsive DEGs of which the vast majority (894 DEGs) overlap with those of males in the GFP plants (*Figure 2A*; *Supplementary file 2A,B*). A higher transcriptional response to females may be expected, because whereas both females and males feed, females also lay eggs on leaves, and plants induce defence responses associated with egg laying of insects (*Stahl et al., 2020*; *Little et al., 2007*). In contrast, leaves of SAP54 plants display dramatically different transcriptional response to the insects, and with 2456 male-specific and only 571 female-specific DEGs, the leaves

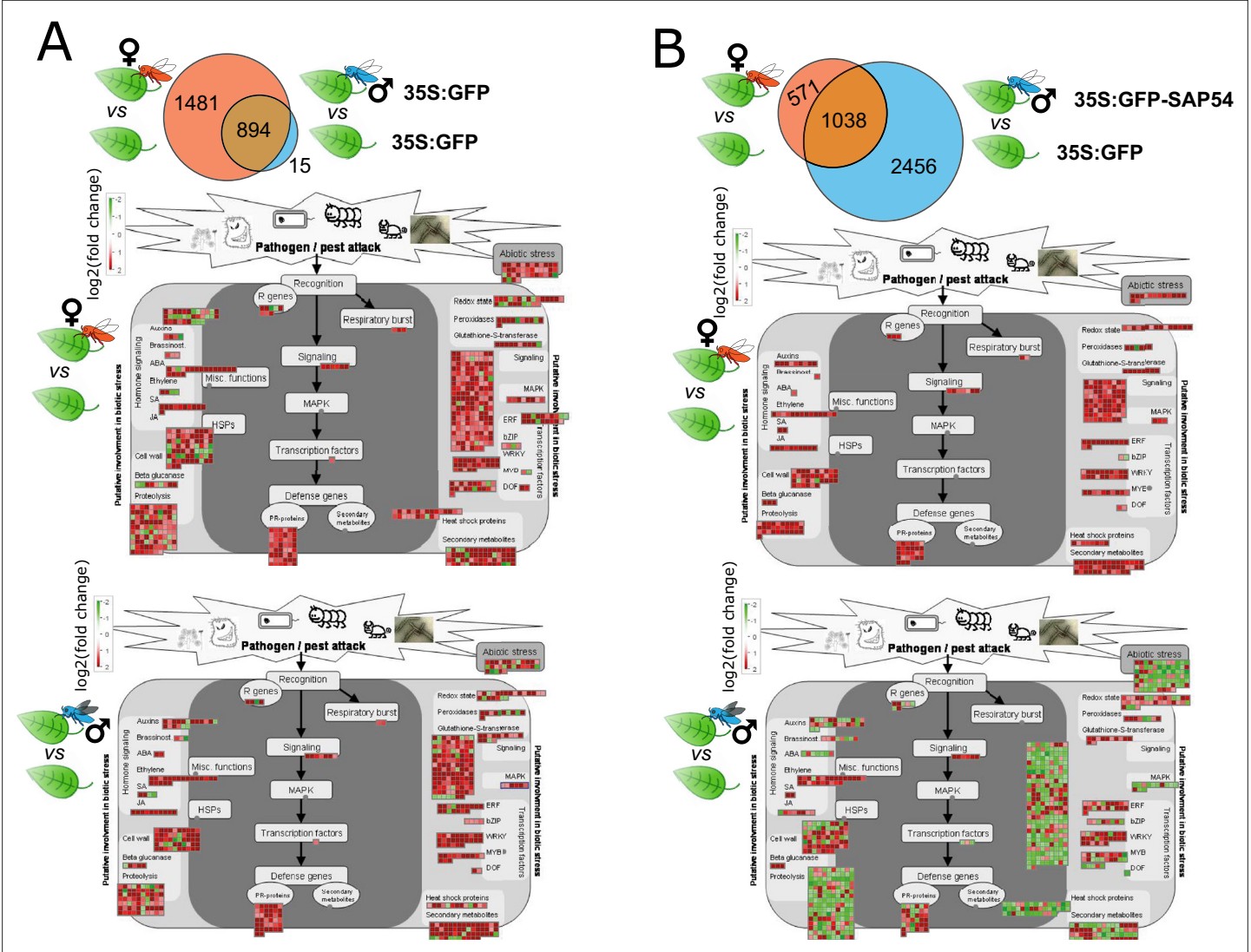

**Figure 2.** SAP54 plants display a dramatically altered leaf response to male leafhoppers by transcriptionally downregulating the majority of biotic stress and plant defence related processes. (**A**-**B**) Euler-Venn diagrams illustrating DEGs in leaves of GFP plants (**A**) and SAP54 plants (**B**) exposed to female leafhoppers compared to male leafhoppers, versus leaves of plants in the control group (cage-only, non-exposed plants). DEG analysis was performed on 17,153 leaf-expressed genes available in *Supplementary file 1*. DEG IDs listed within each Venn diagram are provided in *Supplementary file 2*. (**C**-**D**). MapMan diagrams of *A. thaliana* DEGs involved in biotic stress from female (red insect) or male (blue insect) exposed GFP (**C**) or SAP54 plants (**D**). Biotic stress related pathways were significantly enriched with DEGs from male exposed SAP54 plants compared to other functions listed in *Supplementary file 3*. Names of functional bins (e.g. respiratory burst or MAPK) are listed next to the corresponding color boxes and fully listed in *Supplementary file 4* along with individual transcript names and their fold changes. Red color boxes indicate upregulated, but green – downregulated DEGs based on log₂(fold change).

The online version of this article includes the following figure supplement(s) for figure 2:

**Figure supplement 1.** Experimental design and selection of transcripts for downstream analysis.

**Figure supplement 2.** Biological variation and role of outliers in separation of treatments and identification of differentially expressed genes.

**Figure supplement 3.** The cage-only SAP54 vs cage-only GFP treatments show a limited number of biotic stress DEGs.

particularly exhibit an altered transcriptional response to leafhopper males (*Figure 2B*; *Figure 2— figure supplement 3A*; *Supplementary file 2C,D*). This is striking particularly given that only minimal transcriptional differences (266 DEGs) are observed between leaves of SAP54 and GFP plants that were not exposed to insects (*Figure 2—figure supplement 3A*; *Supplementary file 2E*). These data suggest that SAP54 modulates leaf responses of plants that are challenged by leafhopper herbivory, particularly leafhopper males.

# SAP54 plants exposed to males display significant downregulation of biotic stress responses

To better understand how SAP54 may alter leaf responses to the insects, *A. thaliana* genes were binned into functional groups and these groups were analysed for enrichment of DEGs among treatments using the MapMan built-in Wilcoxon rank sum test with Benjamini-Hochberg correction (*Usadel et al., 2005*). Leaves of GFP plants exposed to females show enrichment for DEGs in 30 functional groups compared to those of non-exposed GFP plants, as opposed to only 2 groups in GFP plants exposed to males (*Supplementary file 3*). However, strikingly, leaves of the SAP54 plants exposed to males show enrichment of DEGs in 42 functional groups compared to those of non-exposed SAP54 plants, as opposed to only 11 groups in leaves of SAP54 plants exposed to females (*Supplementary file 3*). The majority of the functional groups, and particularly in leaves of the male-exposed SAP54 plants, included genes with known functions in biotic stress responses, hormone metabolism, signaling, secondary metabolism and other plant-defence-related processes (*Supplementary file 3*). Moreover, bins enriched for genes in respiration (glycolysis) and cell wall modification functions were uniquely enriched in the leaves of male-exposed SAP54 plants (*Supplementary file 3*). In the no-insect/cage-only SAP54 versus GFP treatments, only 1 functional group (biotic stress responses involving plant defensins) was significantly enriched in the leaves (*Supplementary file 3*) and the majority of genes were upregulated in leaves of the SAP54 plants (*Figure 2—figure supplement 3B*). Therefore, SAP54 alone has a relatively minor impact on leaf responses. Visualization of the fold change of these genes in MapMan graphs focused on genes with roles in plant responses to pathogen and pest attack (*Usadel et al., 2005*) show upregulation of the majority of biotic stress response DEGs in leaves of female- and male-exposed GFP and those of female-exposed SAP54 plants, as opposed to a prominent downregulation of the majority of DEGs in leaves of male-exposed SAP54 plants (*Figure 2C and D*; *Supplementary file 1*). Specifically, multiple genes with roles in abiotic stress, cell wall modification, proteolysis, respiratory burst, secondary metabolism and defence signaling (MAPKs) were downregulated in the male-exposed SAP54 leaves. Together these results indicate that SAP54 dramatically alters leaf biotic stress response to leafhoppers, and downregulates the majority of this response in the presence of males.

We wished to further characterize which components of defence pathways were predominantly affected in male-exposed leaves of SAP54 plants. To achieve this, we performed a de novo manual curation of *A. thaliana* defence signaling module, integrating published literature and public TAIR database annotations for known families of membrane receptors and receptor-like kinases (*Shiu and Bleecker, 2001*), cytoplasmic receptors such as NLR proteins (*Hofberger et al., 2014*; *Kroj et al., 2016*; *Sarris et al., 2016*), CDPK-SnRK superfamily (*Hrabak et al., 2003*), MAP kinases cascade (*Asai et al., 2002*; *Jonak et al., 2002*) as well as known salicylic acid, jasmonic acid and ethylene biosynthesis and signaling genes (*van Verk, 2010*). These manually curated defence signaling modules are provided here as *Figure 3—figure supplement 1* and *Supplementary file 5*, and were imported into the MapMan tool for visualization and DEG analyses (*Figure 3*). Genes for LRR cell surface receptors, CC-NBS-LRRs, RLCKs, MAPKs, JA biosynthesis and ET biosynthesis and signaling were largely down regulated in leaves of the male-exposed SAP54 plants compared those of the other treatments (*Figure 3*). Among the DEG-enriched biotic stress or defence signaling pathway bins, the one for LRR receptors is most enriched and most of these transcripts were downregulated in male-exposed leaves of SAP54 plants compared to the other treatments (*Supplementary file 6*).

Given our finding that females prefer to reproduce on male-exposed SAP54 leaves over those of male-exposed GFP plants in the choice experiments, and don't prefer the female-exposed leaves of SAP54 plants (*Figure 1*), we also compared leaf transcriptional profiles of the male-exposed and female-exposed SAP54 versus GFP treatments. Leaves of SAP54 plants respond more to males (2857 male-specific DEGs) than to females (179 female-specific DEGs) compared to those of GFP plants with 957 DEGs shared between the two exposures (*Figure 4A*). Strikingly, none of the functional categories (bins) are enriched for DEGs in the female exposure treatment, as opposed to 51 categories in the male exposure treatment (*Supplementary file 7*). The 957 shared DEGs do not show enrichment for functional categories, though removing the shared DEGs from the male dataset by considering only the 2857 male-specific DEGs reduced the enriched categories for the male treatment to 41. Among the latter DEG-enriched categories were protein synthesis, signaling involving receptor kinases, cell wall modification, hormone metabolism, including those of jasmonate and ethylene, and secondary

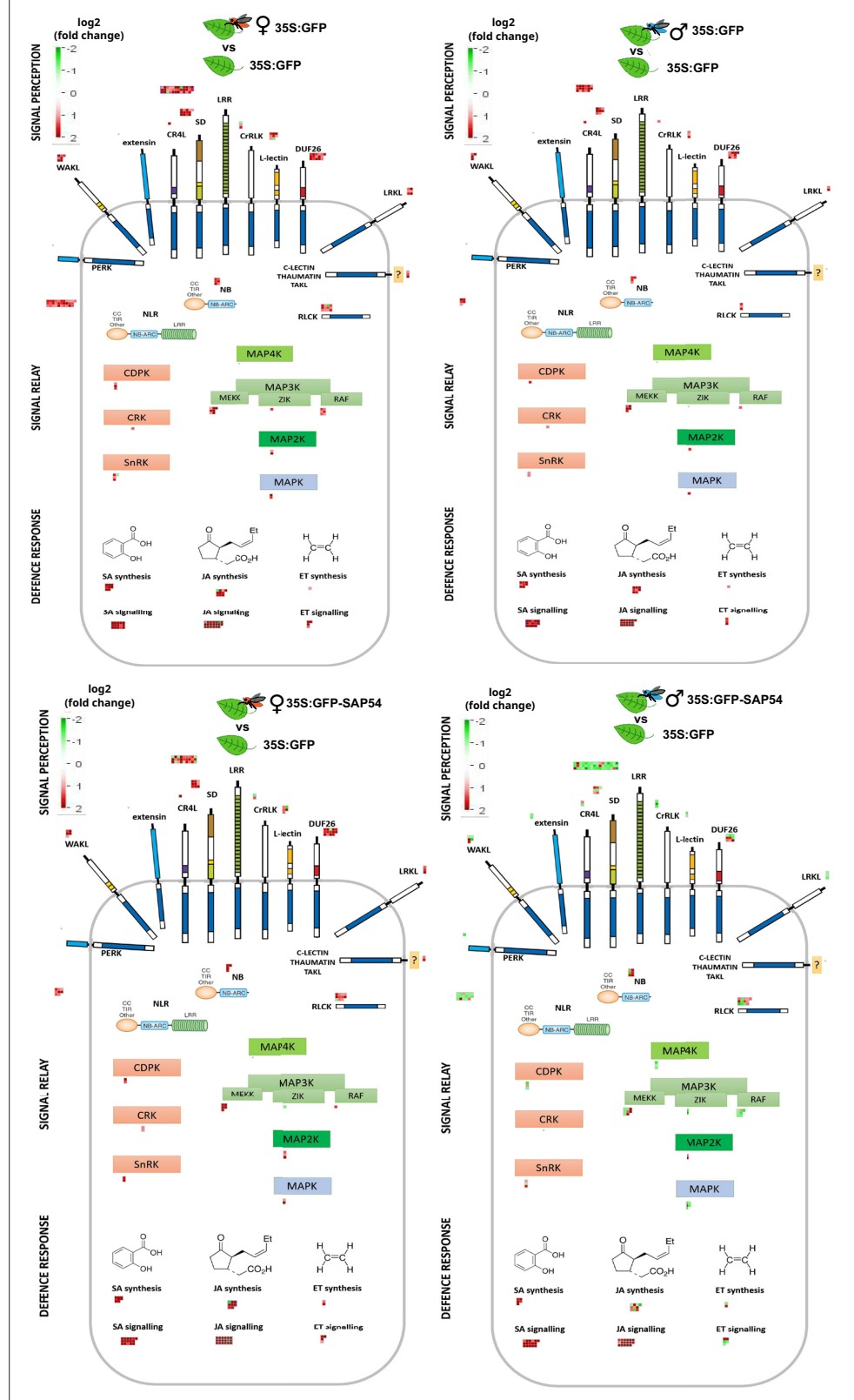

**Figure 3.** Genes involved in *A. thaliana* defence responses are predominantly down regulated in male-exposed leaves of SAP54 plants. MapMan diagrams with the manually curated bins for plant cell-surface receptors, NLRs, RLCKs, MAPKs and hormone biosynthesis and signaling proteins involved in plant responses to biotic stress. Full list of transcripts assigned to each bin can be retrieved from ***Supplementary file 5***. DEGs are indicated as boxes

*Figure 3 continued on next page*

*Figure 3 continued*
above or adjacent to each protein category. Red color boxes indicate upregulated, but green – downregulated DEGs based on log₂(fold change). Female (red insect) or male (blue insect) exposed GFP or SAP54 plants are always compared to insect free plants for DEG analysis and indicated by the insect image above each panel. Individual transcript names and their fold changes within each panel are listed in *Supplementary file 6*. The manually drawn defence signaling background image can be downloaded as *Figure 3—figure supplement 1*.

The online version of this article includes the following figure supplement(s) for figure 3:

**Figure supplement 1.** Manually drawn MapMan image for defence signaling pathway visualization.

metabolism of defence compounds. MapMan graphs of biotic stress responses show downregulation of all 957 shared DEGs and the majority of 2857 male-specific DEGs, and only few downregulated genes in the 179 female-specific DEGs (*Figure 4B*; *Supplementary file 8*). Among the biotic stress response genes, those that are significantly enriched and display the greatest fold-change among male-specific DEGs in leaves of SAP54 plants encode cell surface receptors of LRR and WAKL family and MAP kinases, as well as those involved the ethylene synthesis and signaling, protein degradation and isoprenoid (relating to lignin, glucosinolate, and flavonoid biosynthesis) pathways (*Figure 4B*; *Supplementary file 8*). In summary, protein translational processes and defence pathways predominantly downregulate in male-exposed leaves of SAP54 plants compared to those of GFP plants, and this response was not obvious in the female-exposed leaves. Therefore, leaf defence responses of SAP54 plants are downregulated specifically in response to male exposure and this could explain female preference for the male-exposed SAP54 plants.

## The MADS-box transcription factor SVP is required for female preference of male-exposed SAP54 plants

MADS-box transcription factors (MTFs) are predominantly investigated for their involvement in regulation of flowering. However, how MTFs may affect leaf responses to biotic stress, such as leafhopper colonization, is less well investigated. SAP54 mediates the degradation of many MTFs via the 26 S proteasome by recruiting RAD23 proteins, which are 26 S proteasome shuttle factors, and RAD23 proteins are required for both MTF degradation and leafhopper preference for SAP54 plants (*MacLean et al., 2014*). Therefore, we wished to investigate if the SAP54-mediated degradation of specific MTFs is required for the female preference for male-exposed SAP54 plants. The 17,153 leaf transcripts identified earlier (*Figure 2—figure supplement 1B*) were filtered for the presence of 107 known MTFs (*de Folter et al., 2005*) identifying 20 MTFs that are expressed in *A. thaliana* leaves independent from the presence of SAP54 (*Supplementary file 9*). The majority of the MTFs show differential expression among the male or female-exposed or cage-only leaves of SAP54 and GFP plants, including SVP, SOC1 and several of the MAFs (*Supplementary file 9*). The most striking included MAF5, which showed substantial downregulation in the male-exposed versus cage-only leaf responses of GFP plants and substantial upregulation in leaves that had combined exposure to SAP54 and leafhoppers (*Supplementary file 9*). Others, such SVP and AGL24, were mostly downregulated in the presence of insects or SAP54 and upregulated in the combined presence of insects and SAP54 (*Supplementary file 9*). The MADS-box transcription factor genes had similar expression levels between female-exposed and male-exposed leaves of SAP54 vs GFP plants, except MAF4 that was substantially downregulated in leaves of male-exposed SAP54 vs GFP plants and substantially upregulated in female-exposed SAP54 vs GFP plants (*Supplementary file 9*). Therefore, MADS-box transcription factor genes are differentially expressed in leaves upon exposure to insects and/or presence of SAP54.

We and others previously found that SAP54 directly interacts with many MADS-box transcription factors and mediates their degradation via recruiting RAD23 (*MacLean et al., 2014*; *Kitazawa et al., 2022*; *Suzuki et al., 2024*). Here, we found that SAP54 mediated the degradation of several MADS-box transcription factors for which we observed corresponding genes to be expressed in leaves (*Figure 5A*). SAP54 interacts directly with the majority of these (*Figure 5B*). However, it does not appear to interact directly with SVP (*Figure 5B*). Nonetheless, SVP interacts with other MTFs, including SOC1 and AGL6, which interact with SAP54 (*Figure 5B*) and SVP forms heteromeric complexes with SOC1, FUL, AGL6, AGL24 and several other MTFs (*de Folter et al., 2005*; *Gregis et al., 2006*; *Gregis et al., 2009*; *Balanzà et al., 2014*; *Mateos et al., 2015*). Moreover, SAP54

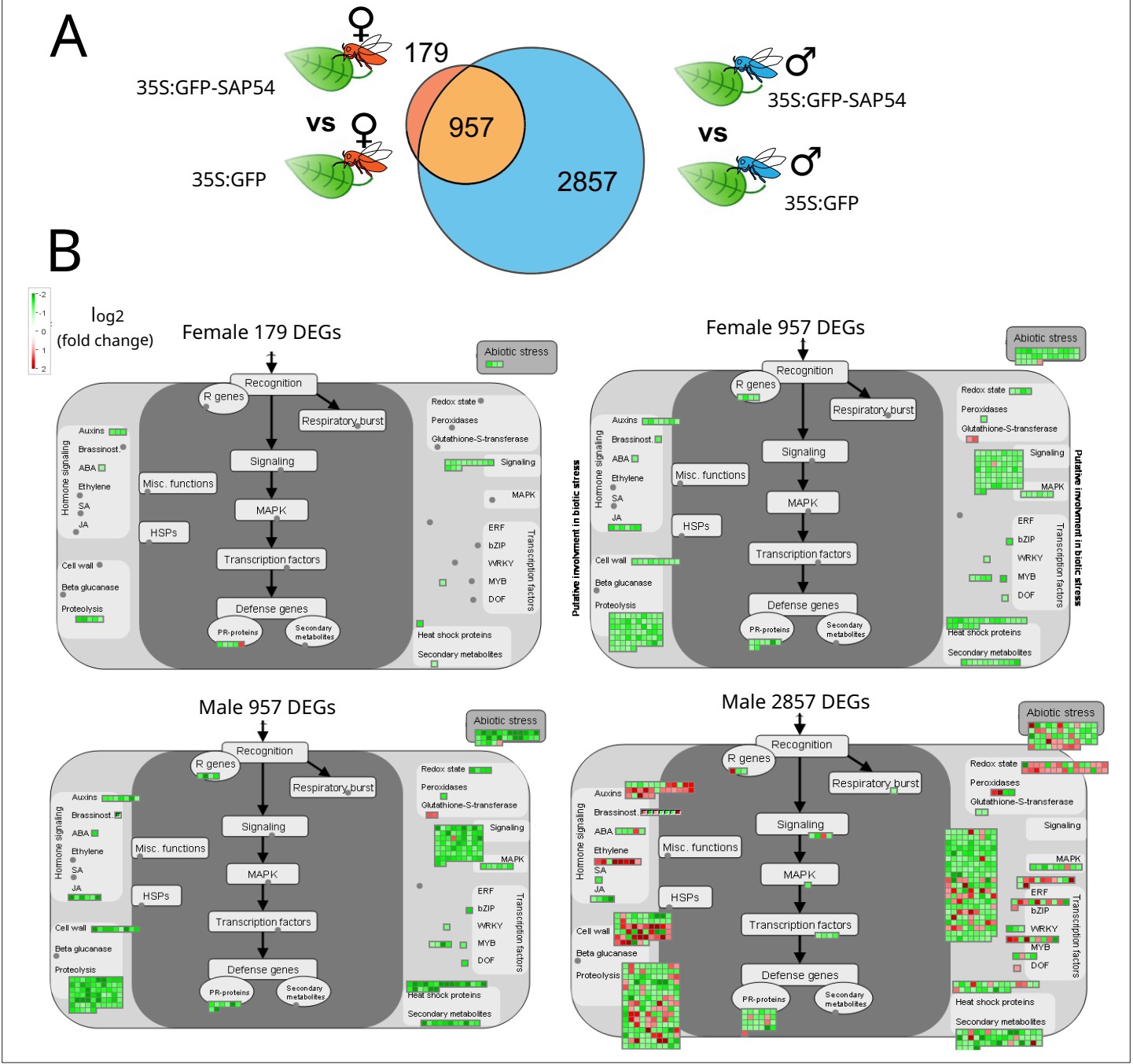

**Figure 4.** Biotic stress response genes are predominantly downregulated in male-exposed SAP54 versus GFP plants. Plant biotic stress is among the most enriched bins with male-specific responses in SAP54 leaves.

mediates degradation of SOC1 and other MTFs via the 26 S proteasome (**MacLean et al., 2014**). Hence, SAP54-mediated destabilization of SVP could occur indirectly via SAP54 interactions with SOC1, AGL6 and other MADS-box transcription factors that form multimeric complexes with SVP.

Next, we conducted leafhopper choice tests with *null* mutants for several MADS-box transcription factor genes. These revealed that the leafhoppers preferred to reproduce on the *svp* and *maf5* mutants, whereas no clear preferences of the leafhoppers were observed for *A. thaliana agl24, sep4, maf1, maf4, ful,* and *soc1* mutants compared to wild-type plants (**Figure 6A**; **Figure 6—figure supplement 1**). Moreover, leafhopper egg-laying preference was abolished when SAP54 was introduced into the *svp* mutant background, but not when introduced into the *maf5* mutant background (**Figure 6B**;

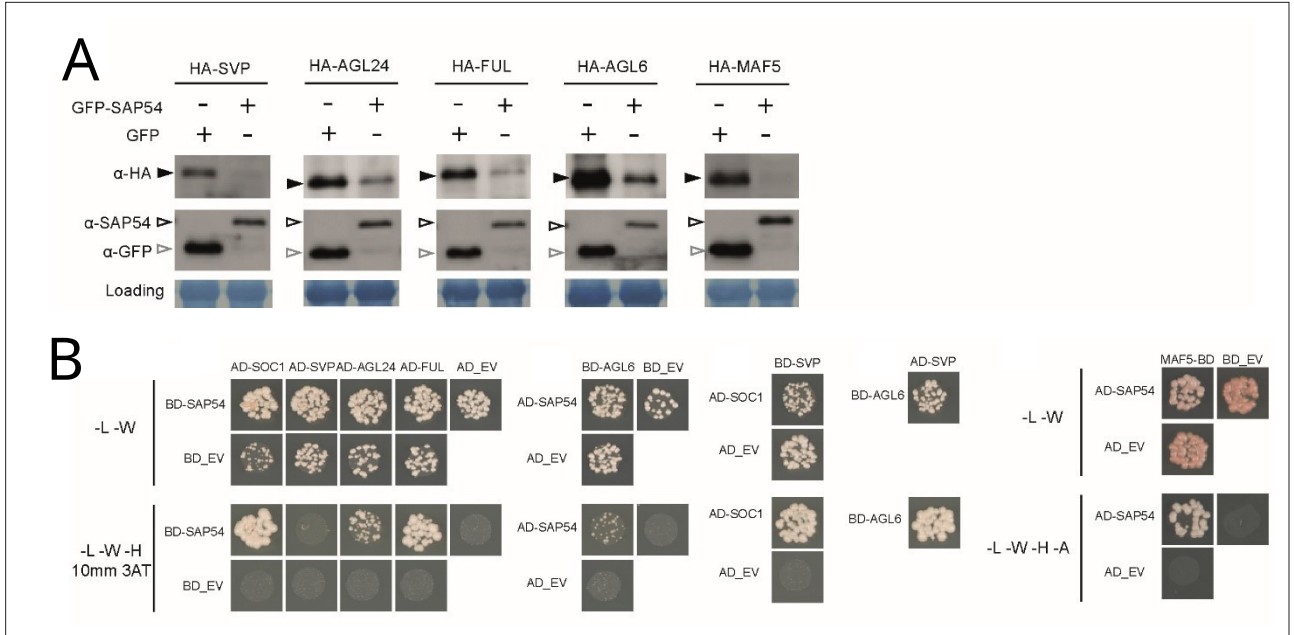

**Figure 5.** SAP54 interacts with multiple MADS-box transcription factors and mediates their degradation. (**A**) Western blots showing degradation of MTFs in the presence of SAP54 in *A. thaliana* protoplasts. Assays were repeated twice with similar results and available in *Figure 5—figure supplement 1* and the associated source data. (**B**) Yeast two-hybrids assays with GAL4-activation (AD) and GAL4-DNA binding (BD) domains fused to the test proteins. -L-W-H-A denote auxotrophic SD media lacking leucine, tryptophan, histidine or adenine, conditionally supplemented with 3-Amino-1,2,4-triazole (3AT). EV, empty vector control.

The online version of this article includes the following source data and figure supplement(s) for figure 5:

**Figure supplement 1.** SAP54 mediates degradation of multiple MADS-box transcription factors.

**Figure supplement 1—source data 1.** Gel/blot source data.

**Figure supplement 1—source data 2.** Gel/blot source data.

*Figure 6—figure supplement 1*), suggesting that that plant SVP is the dominant MADS-box transcription factor required for the SAP54-dependent leafhopper preference. In parallel, choice tests were done with AY-WB-infected wild-type and mutant plants, and the leafhoppers did not display the colonization preference for both *svp* and *maf5* mutants (*Figure 6C*; *Figure 6—figure supplement 1*), indicating that AY-WB-infected wild type and mutant plants are equally attractive to the leafhoppers. While females showed a preference for male-exposed *svp* and *maf5* mutants versus male-exposed wild-type plants (*Figure 6A*) this preference for *svp* and *maf5* mutants was abolished in the absence of leafhopper males (*Figure 5D*; *Figure 6—figure supplement 1*). Transcripts for *SVP* and *MAF5* were detected, and upregulated, in leaves in the combined presence of SAP54 and leafhoppers (*Figure 5E*; *Supplementary file 9*), consistent with previous findings of *Yang et al., 2015* showing upregulation of *SVP* in *Catharanthus roseus* by peanut witches' broom phytoplasma effector PHYL1. Together, these data provide evidence that female preference for male-exposed SAP54 plants is dependent upon the presence of SVP, while MAF5 plays a role in female choice independently from SAP54.

## SVP differentially regulates plant responses to herbivores

Considering the role of SVP in leafhopper reproduction preference, we conducted transcriptional analyses to assess how leaves of *svp* mutant plants respond to male and female leafhopper exposure and compared this to male or female exposed SAP54 plants (*Figure 6—figure supplement 2A*). A comparable number of DEGs was found in leaves exposed to both males and females, with 464 DEGs uniquely responding to females, 347 to males, and 460 to both sexes (*Figure 6—figure supplement 2B*). Many of biotic stress-related DEGs in male or female exposed *svp* mutant were upregulated (*Figure 6—figure supplement 2C*). When searching for functions that were enriched with the 924 DEGs in the female exposed *svp* mutant versus female exposed wild-type plants, we found significant enrichment for abiotic stress and cell wall genes (*Supplementary file 10*) as well as DUF26 family

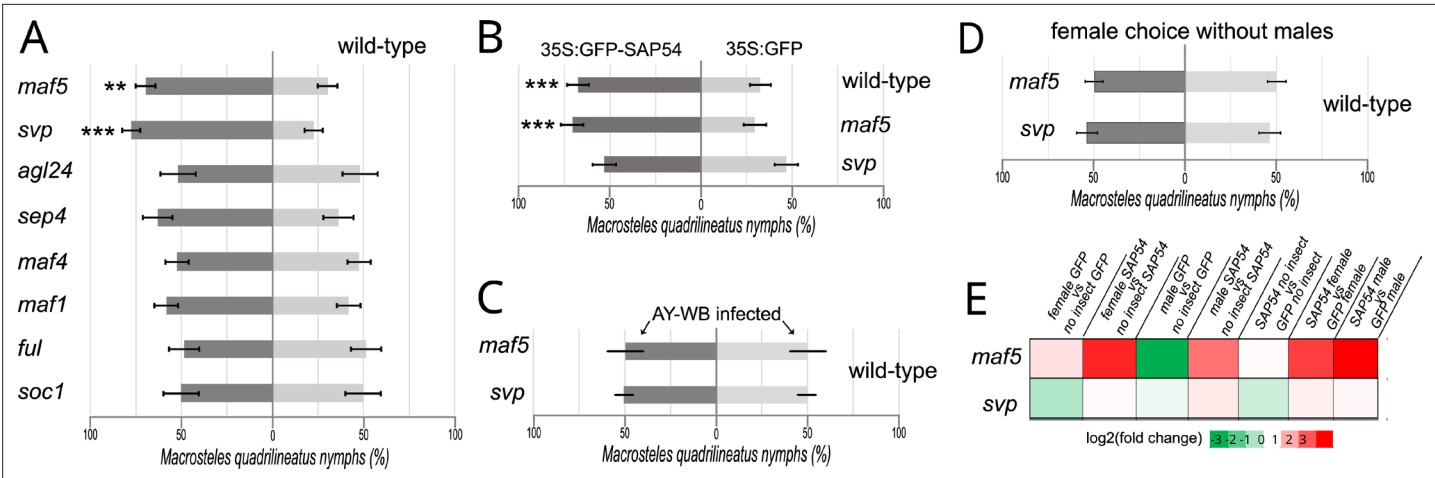

**Figure 6.** The MADS-box transcription factor SVP is required for female preference to reproduce on male-exposed SAP54 plants. (**A-C**) Choice test with equal numbers of 10 males and 10 females on wild type and MTF *null* mutant *A. thaliana* rosettes (**A**), 35 S:GFP-SAP54 *A. thaliana* rosettes (panel B) and AY-WB phytoplasma-infected plants (**C**). (**D**). Choice tests with 10 females on *A. thaliana svp* and *maf5 null* mutants without males. The entire series of choice tests depicted in panels (**A–D**) were conducted in parallel and repeated independently two times. ** p<0.01, *** p<0.001. Bars are ± 1 SEM. (**E**). Expression levels of *SVP and MAF5* in leaves among treatments showing that SVP and MAF5 are upregulated in male-exposed SAP54 plants.

The online version of this article includes the following figure supplement(s) for figure 6:

**Figure supplement 1.** SVP enhances female egg-laying preference for male exposed plants in phytoplasma and SAP54-dependent manner.

**Figure supplement 2.** Comparison between male and female colonized *svp* plants reveal SVP-dependent insect regulated biotic stress genes.

cell surface receptors (*Supplementary file 11*). Furthermore, the 807 DEGs in *svp* leaves exposed to males compared to male colonized wild-type plants were predominantly involved in sulphur-containing glucosinolates and cell-wall-associated proteins (*Supplementary file 10 and 11*). Hence, the DEGs linked to cell wall structure and modulation were prominently enriched in leaves exposed to both male and female leafhoppers. There was no clear enrichment of functional categories among the 347 male-specific DEGs in *svp* mutant (*Supplementary file 10 and 11*). These findings suggest that SVP influences leaf responses to leaf-colonizing insects in a manner that is partially specific to the gender of the leafhoppers.

We further wished to explore the DEGs that are common between the male-exposed leaves of SAP54 and *svp* mutant plants. Firstly, we analysed the overlap between male-specific (not up- or down-regulated by female) SAP54 expressing plants and the *svp* mutant, where we found only 49 DEGs (*Figure 6—figure supplement 2D*). Many of these genes display similar up- or down-regulation in SAP54 and *svp* plants, and some of the most downregulated DEGs include cytochrome P450s such as CYP81F, CYP78A as well as glutamate receptor GLR3 and calcium-binding transcription factor MYC5 (NIG1; *Supplementary file 12A*).

However, due to the relatively small number (49) of shared DEGs for full pathway characterization, we analysed functions encoded by all 155 shared DEGs in male exposed SAP54 and *svp* plants, including those that overlap with the female regulated DEGs in those plants (*Figure 6—figure supplement 2E*). The 155 DEGs showed significant GO term enrichment associated with various biological responses, including responses to stimuli, oxygen-containing compounds, lipids, hormones, and calcium signaling (*Supplementary file 12B*), suggesting common functions in biotic stress regulation between SAP54 and SVP in male exposed plants. Moreover, we also examined the full spectrum of MapMan biological functions enriched by DEGs in leaves of male-exposed SAP54 (3814 DEGs) and *svp* mutant (807 DEGs) plants separately before looking for the overlap in functions (rather than gene identities first). This comparison revealed that the similarities between the enriched functions in male exposed SAP54 and *svp* mutant plants predominantly pertain to cell wall modification and secondary (isoprenoid) metabolism (*Supplementary file 12C*). Considering that female insects are more attracted to the male-exposed leaves of SAP54 and *svp* mutant plants for feeding and oviposition, alterations in cell wall structure and changes in secondary (isoprenoid) metabolism may potentially play significant roles in influencing female preferences.

## Discussion

Here, we investigated colonization preferences of the leafhopper vector *M. quadrilineatus*, the predominant vector of AY-WB phytoplasma, for plants that produce the AY-WB effector SAP54. We made the surprising discovery that female preference for SAP54 plants only occurs in the presence of males. Both SAP54 and males are necessary for this preference, as females did not show a preference in their absence. As well, SAP54 plants exposed to males display significant downregulation of biotic stress responses. We identified the MADS-box transcription factor SVP to be essential for this female preference to male-exposed SAP54 plants. Leafhopper females exhibited a preference for reproducing on *svp null* mutant plants, a behavior that is also contingent upon the presence of leafhopper males, and aligning with our observation that SAP54 mediates the degradation of SVP in leaves. Therefore, by characterizing a parasite effector, we revealed the crucial role of SVP in leaf susceptibility to male herbivorous insects, enhancing female attraction and colonization.

This work built on our previous findings that the leaf-like flowers are not required for the insect colonization preference and the insect vectors are predominantly attracted to the leaves and not flowers of plants (*Orlovskis and Hogenhout, 2016*). Here, we independently confirm these data via experiments that show a preference of the insects for *A. thaliana* rosettes before the onset of flowering. Therefore, a morphological change in the host may not always be responsble for the extended phenotype effect on the alternative host, which is in this case the insect vector. It is possible that the attraction of insect vectors to leaves is the primary function of SAP54, and the induction of the leaf-like flowers an evolutionary side-effect of SAP54 adaptation to target MTFs that leads to degradation of SVP. However, the prevention of seed production through the formation of leaf-like flowers likely is likely to present an advantage to phytoplasma. AY-WB phytoplasma commonly infects annual plants, where leaves naturally senesce and die off after flowering and seed production. It is probable that prolonging the longevity of their plant hosts is advantageous for phytoplasmas, particularly considering that they are not known to be seed-transmitted. Instead, phytoplasmas rely on leafhoppers, which feed on vegetative tissues, for transmission.

We found that SAP54 mediates the destabilization of SVP. However, we did not find evidence of a direct interaction between SAP54 and SVP in our Y2H analyses. An explanation of this apparent discrepancy is that MTF proteins exhibit a propensity to form diverse complexes, including homo- and hetero-dimers, as well as higher order ternary and quaternary complexes (*Davies et al., 1996*; *Egea-Cortines et al., 1999*; *Hartmann et al., 2000*; *Honma and Goto, 2001*; *Immink et al., 2002*; *de Folter et al., 2005*). Notably, SVP is known to interact with other MTFs such as SOC1, FUL, and AGL24 (*de Folter et al., 2005*; *Gregis et al., 2009*; *Balanzà et al., 2014*), and we previously established that SAP54 interacts with and facilitates the degradation of multiple MTFs, including SOC1, FUL, and AGL24 (*MacLean et al., 2014*), as is also shown for FUL and AGL24 herein (*Figure 5*). Moreover, our research showed that these three MTFs are expressed in leaves, with SOC1 and AGL24 showing elevated expression in SAP54 plants exposed to males (*Supplementary file 9*). Hence, the destabilization of SVP by SAP54 may occur through its involvement with various SVP-containing protein complexes. This intricate interaction network also provides a possible explanation for the absence of a leafhopper preference phenotype in *soc1*, *agl24*, and *ful* mutant plants. It suggests that the interaction of SAP54 with multiple SVP-containing ternary and quaternary MTFs, leading to their degradation, might be essential to trigger the observed phenotype.

SVP is recognized as a conserved regulator of flowering time across a variety of eudicot species, as evidenced by several studies (*Sun et al., 2016*; *Li et al., 2016*). It is expressed in the leaves during the vegetative growth stages of plants (*Zheng et al., 2009*; *Li et al., 2016*; *Wu et al., 2017*), which correlates with the wide host range of AY-WB phytoplasmas. While the primary role of SVP and its homologs has been associated with flowering, their involvement in additional biological processes beyond this remains underexplored (*Liu et al., 2018*). However, emerging research has begun to shed light on the roles of SVP outside of flowering: for instance, it acts as a positive regulator of age-related resistance against bacterial pathogens in *Arabidopsis thaliana* (*Wilson et al., 2017*) and serves as a negative regulator of various Jasmonate Zim-domain (JAZ) genes during the vegetative growth phase (*Gregis et al., 2013*). Our findings contribute to this expanding body of knowledge, suggesting that plant immunity may be intricately controlled by developmental regulators like SVP, which possesses multiple, diverse functions throughout plant development (*Berry and Argueso, 2022* references therein). The observation that female leafhoppers prefer to reproduce on both SAP54 and *svp* mutant

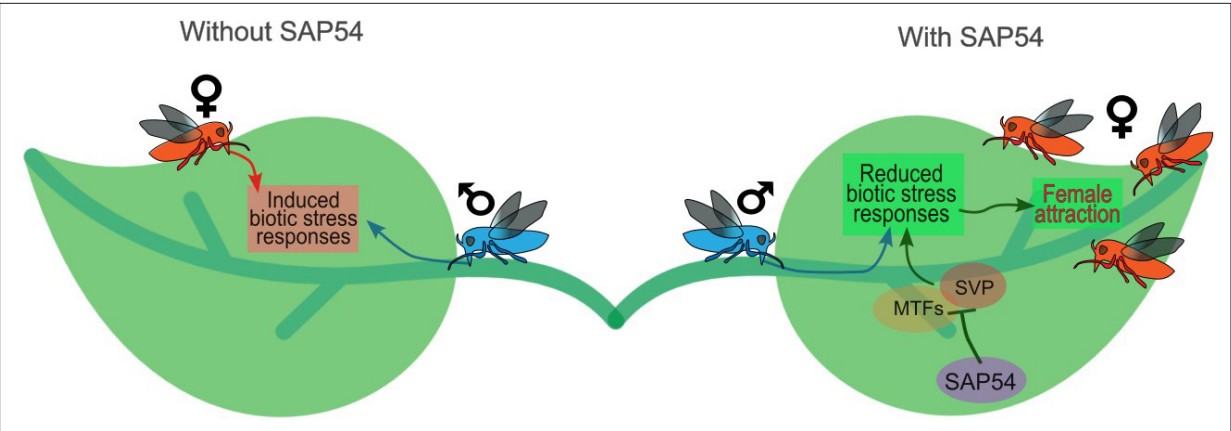

**Figure 7.** Model illustrating how the phytoplasma effector SAP54 facilitates leaf colonization by leafhopper vectors, which are essential for phytoplasma transmission and spread. During phytoplasma infection, the SAP54 effector is secreted and promotes the degradation of MADS-box transcription factors. This degradation of MADS-box transcription factor short vegetative phase (SVP) specifically leads to the downregulation of biotic stress responses to male leafhoppers, thereby attracting female leafhoppers to the leaves. The females then lay eggs, and their offspring acquire phytoplasmas during feeding. Thus, SAP54 likely enhances phytoplasma spread by increasing female insect vector reproduction in a process that requires SVP and the presence of males. An additional implication of this model is that male leafhoppers might benefit from phytoplasma infection by attracting more mates. This work corroborates the previous findings that both the SAP54-mediated degradation of MADS-box transcription factors and the leafhopper attraction phenotype are dependent on the 26 S proteasome shuttle factor RAD23 (*MacLean et al., 2014*) and that leafhoppers are specifically attracted to leaves and not the leaf-like flowers that are induced by SAP54 actions (*Orlovskis and Hogenhout, 2016*).

plants exclusively in the presence of leafhopper males underscores a sophisticated regulatory effect of SVP and other MADS-box transcription factors targeted by SAP54 on plant responses to the different sexes of the herbivores. This intricate interaction is detailed in the mechanism proposed in *Figure 7*.

Considering that certain phytoplasmas can modify plant volatile production (*Mayer et al., 2008a*; *Mayer et al., 2008b*; *Rid et al., 2016*; *Tan et al., 2016*), we examined whether male-induced volatile cues play a role in attracting females. Our results suggest that female leafhopper preference for laying eggs on male-exposed SAP54 plants is unlikely to depend solely on volatiles originating from the males or induced by them (*Figure 1—figure supplement 3*). Thus, it is plausible that females are guided to male-occupied SAP54 and *svp* mutant plants through a different mechanism, possibly involving changes in leaf responses that are specifically induced to male insects. Male-exposed leaves of SAP54 and *svp* mutant plants shared transcriptional changes of genes involved cell wall structure processes and changes in secondary (isoprenoid) metabolism. As such, these could influence female leafhoppers' host plant selection and their egg-laying preferences on plants exposed to males. However, given that the removal of males from SAP54 leaves prior to female choice does not enhance female colonization (comparison of *Figure 1* treatment 4 with treatment 5), we cannot exclude that male produced volatiles or mating calls can enhance or supplement SAP54-dependent changes in biotic stress responses to males for enhanced female attraction.

Males and females of *M. quadrilineatus*, the primary vector of AY-WB phytoplasma, exhibit distinct movement and feeding behaviors, as documented by *Beanland et al., 1999*. Moreover, the leafhoppers, especially females, live longer on aster yellows infected plants compared to non-infected plants (*Beanland et al., 2000*). Here, we show that the combination of the aster yellows phytoplasma effector SAP54 and males attract females in a manner that depends on SVP. Moreover, plant biotic stress and defence responses are largely suppressed in male (compared to female) exposed SAP54 plants, and SVP also displays distinct plant transcriptional responses to male and female leafhoppers. Nevertheless, the mechanisms how male and female leafhoppers, in combination with SAP54, differentially affect plant defence responses will need further investigation. While the differences in male and female behavior may play a role (*Beanland et al., 1999*), there is also the possibility that effectors in the saliva of sap-feeding insects modulate plant processes, including defence (*Snoeck et al., 2022*), and sex-specific differences exist in the proteome of sap-feeding insect saliva (*Miao et al., 2018*), potentially influencing plant response. Moreover, plant immune responses initiated by the molecular patterns associated with insect eggs (EAMPs) that bear similarities to pattern-triggered immunity (PTI)

activated by bacterial pathogens (*Gouhier-Darimont et al., 2013*) may play a role in the observed gender-specific differences.

Pathogens uniquely exploit sexual dimorphism in their hosts to enhance their dispersal and survival in nature. For example, the bacterial parasite infecting the crustacean *Daphnia magna* selectively uses male hosts for horizontal transmission, while exploiting females for internal replication and spore production, thus facilitating widespread disease transmission (*Nørgaard et al., 2019*). Toll signaling pathway contributes to the varied resistance observed between male and female *Drosophila* to both Gram-positive and Gram-negative bacteria (*Duneau et al., 2017*). Similarly, the endosymbiotic bacterium *Wolbachia* often exhibits distinct effects on the embryonic development of male and female gametes, showcasing another instance of pathogens leveraging host sexual differences (*Serbus et al., 2008*). Recent discovery identified a novel *Wolbachia* factor Oscar which interacts with host protein Masculinizer to inhibit masculinization and lead to male-killing in two insect species (*Katsuma et al., 2022*). Our study further expands the mechanistic understanding about insect gender specific factors targeted by parasites.

In our research, we have uncovered a mechanism whereby the phytoplasma effector SAP54 targets a conserved host regulator, SVP, to attract female leafhopper vectors to produce more progeny on plants exposed to leafhopper males. Given that phytoplasmas rely on leafhoppers for acquisition during feeding and subsequent transmission to other plants, SAP54 likely promotes the abundance of phytoplasma-carrying insect vectors and facilitates phytoplasma transmission. This illustrates how a bacterial gene extends its influence beyond its primary plant host to modulate the host choice and reproductive behavior of the insect vector. Such findings underscore the complexity of tripartite plant-insect-microbe interactions, wherein parasitic genes exhibit what could be termed 'hyper-extended phenotypes,' manipulating the biology of one host to facilitate transmission through another host. This research advances our understanding of plant-microbe interactions and resonates with the concept of the Extended Evolutionary Synthesis (EES), as articulated by *Hunter, 2018*. Within the EES framework, parasitic genes like SAP54 evolve adaptively through both natural selection and direct biochemical interactions with plant regulators, as well as through the extended functions these regulators serve in interactions with insect vectors. In this study, we found that the extended role includes enhancing the likelihood of male vectors to attract females for reproduction.

## Materials and methods
### Generation of plants for insect choice experiments
Generation of 35 S:GFP-SAP54 and 35 S:GFP transgenic *Arabidopsis* lines was done according to methods described in *MacLean et al., 2011*; *MacLean et al., 2014*. *ful-1* line was provided by Lars Ostergaard Lab and described in *Gu et al., 1998*. *soc-1* and *sep4-1* by Richard Immink Lab and described in *Immink et al., 2012*. *maf4-2 and maf5-3* seeds were provided by Hao You Lab and described in *Shen et al., 2014*. *maf1* (also known as *flm-3*) was provided by Claus Schwechheimer Lab and described in *Lutz et al., 2015*. *svp1-41* was provided by Martin Karter Lab and described in *Hartmann et al., 2000*. *agl24* (N595007, SALK_095007) was obtained from The Nottingham *Arabidopsis* Stock Centre (NASC). Phytoplasma-infected plants were generated identical to methods described in *MacLean et al., 2014*. All plants were grown under short days (10 hr/14 hr light/dark) identical to methods described in *Orlovskis and Hogenhout, 2016*.

### Insect oviposition choice assays
All insect choice experiments were performed with a mixture of virgin and pre-mated females identical to methods described in *Orlovskis and Hogenhout, 2016* with the following modifications. Female-only oviposition choice experiments described in *Figure 1* (*experiment 2*) were set up by releasing 10 *M. quadrilineatus* females, while the mixed-sex choice assay (experiment 1) was done with 10 male and 10 female insects. Five male leafhoppers were confined on test and another five on control plants using transparent clip-cage (2 cm diameter) before the release of 10 females in *experiments 3* and *4* displayed in *Figure 1*. For the *experiment 5* in *Figure 1*, 5 females were restricted on test and 5 on control plants prior the release of another 10 females in the choice cage. Notably, single-sex insects were selected from mixed-sex population of *M. quadrilineatus* stock. Adults were removed 5 d after release, nymphs counted 14 d after adult removal. Nymph counts analyzed using paired t-test.

## Insect olfactory choice assay

Test and control plants were placed in a black (non-transparent) plastic container 12cm x 10cm x 10 cm (H x W x D) with a perforated topside, permitting diffusion of plant odors but rendering plants invisible from the outside of the container. A colorless or a colored sticky landing platform (OECO, Kimpton, UK) was fitted on the top of each container but not covering the perforations. 20 *M. quadrilineatus* females were released in the center of the arena (transparent polycarbonate 62cm x 30cm x 41 cm [H x W x D]) equal distance from the test and the control containers. When coming in contact with the sticky landing platform, insects stuck permanently to its surface. Within the following 1 hr under the ambient room light the majority of insects made their first landing choice, and were counted. Each experiment utilized clean choice arena, new containers and landing platforms to avoid any possible residual semiochemicals left by insects from the previous experiment which could bias the leafhopper choice in the next experiment. In selected experiments, male leafhoppers were contained in clip-cages on the test plants placed inside the containers.

## Western blotting

Proteins were separated on 12.5% (w/v) SDS-PAGE gels and transferred to 0.45 mm Protran BA85 nitrocellulose membranes (Whatman, UK) using the Bio-Rad minigel and blotting systems following standard protocols. Blotted membranes were incubated in blocking buffer (5% w/v) milk powder in 1 X phosphate buffered saline and 0.1% (v/v) Tween-20 with primary antibody at 4 °C overnight. Anti-SAP54 antibodies were raised in rabbits (Genscript) injected with purified 66His-tagged SAP54. A 1:2000 dilution of anti-SAP54 was used in western blots. Monoclonal mouse anti-HA, anti-GFP, anti-bodies (H3663, SAB4200681, Sigma-Aldrich) were used at a 1:10,000 dilution. Peroxidase-conjugated anti-rabbit or anti-mouse secondary antibody (Sigma-Aldrich) was added to washed blots at a 1:10,000 dilution and incubated at room temperature for 4 hr. Bound antibodies were detected using Immobilon Western Chemiluminescent HRP Substrate (Millipore, UK) when exposed to Super RX film (Fuji-film, Germany). Protein loading was visualized using Ponceau S solution (0.1% (w/v) in 5% acetic acid).

## Yeast-two-hybrid assays

The Matchmaker Gold Yeast Two-Hybrid System (Clonetech) and the DUALhybrid system (Dualsystems Biotech) were used for detecting protein-protein interactions in yeasts. Yeast transformation were carried out following manufacturers' instructions. Successful transformants grew on medium lacking leucine and tryptophan (-L -W). Medium lacking leucine, tryptophan, histidine and Adenine (-L-W-H-A) or Medium lacking leucine, tryptophan, histidine and supplemented with 10 mM 3-AT (-L-W-H+10 mM 3-AT) were used to examine protein-protein interactions. The growth of yeast colonies was scored after 5 days of incubation at 28 °C. Protoplast assays were carried out according to *Yoo et al., 2007*.

## Protoplast assays

Transient expression protoplast assays were performed identical to methods in *Huang et al., 2021*. Mesophyll protoplasts isolated from 4 weeks-old *Arabidopsis* leaves were resuspended at $4 \times 105$ mL$-1$ in MMG solution. 300 µL protoplast solution were then transfected with 24 µg plasmids (12 µg each for double transformation) using PEG–calcium-mediated transfection. pUGW15 plasmid was used for expressing HA-tagged genes, pB7WGF2 - for expressing GFP-tagged genes. pEarleyGate 202-GFP was used to express GFP. Protoplasts were kept at 22 °C for 16 hr in dark and then harvested for western blot analysis.

## Generation of plants for RNA-sequencing

The experiment used 8 weeks old *Arabidopsis thaliana* (Col-0) plants ectopically expressing Aster Yellows phytoplasma strain Withes' Broom effector SAP54 (35 S:GFP-SAP54) or a control construct (35 S:GFP). Plants were grown at short day photoperiod (10 hr/14 hr light/dark). Single fully expanded leaf of each plant was exposed to either five male or five female insects by placing them in a transparent 2 cm diameter clip-cage which confines the leafhoppers to the leaf and prevents from escaping. An empty clip-cage was placed on a leaf for no-insect control. In order to prevent any insects from the room to land on the experimental plants and confound the results, we placed all plants with clip-caged female leafhoppers in a sealed larger cage separate from plants with clip-caged males or plants

with empty clip-cages. Thus, all replicate plants with male, female or no-insect exposed leaves were maintained in similar micro-environment. The experiment with clip-caged males, females and empty clip-cages was done at the same time on the same shelf within the growth room to ensure similar conditions across all treatments. Plant tissue samples were collected 48 hr after exposure to insects and placed straight into liquid $N_2$. Insect number and exposure time of 48 hr was previously experimentally optimized by measuring number of feeding sites and eggs laid per clip-cage. The experiment with clip-caged male or female plants on *svp* mutant and *wild-type* Col-0 plants was performed at a different time than the experiment with 35 S:GFP-SAP54 and 35 S:GFP plants but with identical setup.

## Total RNA extraction for RNA-seq and quality control

The collected leaf area enclosed by the clip-cage (~2 cm diameter) was stored at –80 °C for subsequent RNA extraction using QIAGEN Plant RNeasy kit following the manufacturer's instructions. Initial RNA integrity was assessed by gel electrophoresis (1% Agarose) visualization of ribosomal bands in the extracted dsRNA as well as ssRNA following 65 °C denaturation and immediate transfer on ice to prevent hybridization. Total RNA concentration and quality was assessed using Nanodrop (Thermo Fisher). Total ≥2 µg of each RNA sample at ≥50 ng/µL concentration, 260/280 ratio between 1.9 and 2.1, and 260/230 ratio between 1.5 and 2.0 was submitted for library preparation and RNA-sequencing.

## RNA sequencing, read alignment, and differential expression analysis

Experimental design consisted of four biological replicates for male- or female-exposed or insect-free transgenic 35 S:GFP-SAP54 plants and transgenic 35 S:GFP control plants as well as three biological replicates for *svp* null mutant and wild-type col-0 control plants. cDNA library construction for SAP54 and GFP was performed according to the IlluminaTruSeq protocol, followed by sequencing on Illumina HiSeq2000 platform and pooling four randomized libraries per lane, with 50 bp single-end reads and 25 M read coverage per sample, whereas *svp* null mutant and wild-type libraries were prepared with paired-end reads. Reads were trimmed for low quality and adapter contamination using Trim Galore! v0.4.0 with default settings (*Krueger, 2012*). Trimmed reads were aligned to *A. thaliana* reference genome (TAIR10) by HISat2 (*Pertea et al., 2016*). We identified differentially expressed transcripts with DESeq2 package in R (version 1.2.10; *Love et al., 2014*) using expression count generated by Kallisto v0.42.3 (*Bray et al., 2016*). Independent filtering was employed in in DESeq2 to remove very low expressing transcripts on the basis of normalized counts. Genes were considered differentially expressed if they had a *p* value less than 0.05 after accounting for a 5% FDR according to the Benjamini-Hochberg procedure and if a fold-change in expression of at least 2 was observed.

## Functional analysis of transcriptome data

Enrichment of GO biological functions and GO molecular functions for DEGs were calculated using Fisher exact test (PANTHER) (http://geneontology.org/docs/go-enrichment-analysis). Graphical visualization for functional catagorization of differentially regulated transcripts was performed using MapMan 3.5.1R2 functional annotation tool (*Thimm et al., 2004*). The tool, supporting resources and annotation database available from http://mapman.gabipd.org/web/guest/home. Statistical analysis on enrichment of differentially changed pathways was performed using Wilcoxon rank test and Benjamini-Hochberg (BH) p-value correction (*Usadel et al., 2005*). Methods for manual creation of new pathways and categorical filtering of expression data were adopted from *Usadel et al., 2009* and explained in sections below. The graphical overview of defence signaling cascades in *Figure 3* and *Figure 3—figure supplement 1* were manually drawn and loaded as separate pathway image files into MapMan. The mapping file (*Supplementary file 5*) containing a list of gene identifiers and descriptions manually assigned to new functional bins previously absent from the MapMan annotation was generated from published literature and the public TAIR database (https://www.arabidopsis.org/browse/genefamily) for membrane-located receptor-like kinases and their classes (*Shiu and Bleecker, 2001*), cytoplasmic receptors such as NLR proteins (*Hofberger et al., 2014*; *Kroj et al., 2016*; *Sarris et al., 2016*), CDPK-SnRK superfamily (*Hrabak et al., 2003*), MAP kinases cascade (*Asai et al., 2002*; *Jonak et al., 2002*) as well as SA, JA and ET biosynthesis and signaling genes (*van Verk, 2010*).

## Materials availability statement

35 S:GFP-SAP54 and 35 S:GFP transgenic *Arabidopsis* lines in MTF-mutant backgrounds were generated for this study and available upon contacting Hogenhout group.

## Acknowledgements

We thank Lars Ostergaard (JIC) for providing *ful-1* seeds, Richard Immink (University of Wageningen) for giving *soc-1* and *sep4-1* seeds, Hao Yu (Zhejiang University) for *maf4-2* and *maf5-3* seeds, Claus Schwechheimer for *maf1* seeds (Technical University of Munich), and Martin Kater (University of Milan) for *svp1-41* seeds. We thank the JIC Entomology and Insectary Platform staff for maintaining the leafhopper and phytoplasma stocks and the JIC Horticultural services staff for growth and maintenance of the plants used in this study. We also thank Sam Mugford (JIC) for technical assistance throughout the project. Allyson MacLean (now University of Ottawa), Vera Thole, Enrico Coen (JIC) deserve personal gratitude for their advice and discussions as members of Z Orlovskis PhD thesis supervisory team. We are grateful to Robert Sablowski (JIC) and Sebastian Schornack (University of Cambridge) for constructive feedback on thesis work that helped to conceptualize the study herein. This work was supported by Human Frontier Science Program grant RGP0024/2015 (to SH), the European Research Council and UK Research and Innovation (UKRI) Engineering and Physical Sciences Research Council grant EP/X024415/1 (to SH), BBSRC (BBS/E/J/000PR9797 and BBS/E/JI/230001B), and a BBSRC student fellowship (to ZO).

## Additional information

### Funding

| Funder | Grant reference number | Author |
|---|---|---|
| Human Frontier Science Program | RGP0024/2015 | Saskia A Hogenhout |
| UK Research and Innovation | EP/X024415/1 | Saskia A Hogenhout |
| Biotechnology and Biological Sciences Research Council | BBS/E/J/000PR9797 | Saskia A Hogenhout |
| Biotechnology and Biological Sciences Research Council | BBS/E/JI/230001B | Saskia A Hogenhout |
| Biotechnology and Biological Sciences Research Council | DTP Student grant | Zigmunds Orlovskis |

The funders had no role in study design, data collection and interpretation, or the decision to submit the work for publication.

### Author contributions

Zigmunds Orlovskis, Conceptualization, Data curation, Formal analysis, Validation, Investigation, Visualization, Methodology, Writing – original draft, Writing – review and editing; Archana Singh, Data curation, Investigation, Writing – review and editing; Adi Kliot, Validation, Investigation, Writing – review and editing; Weijie Huang, Investigation, Visualization, Writing – review and editing; Saskia A Hogenhout, Conceptualization, Formal analysis, Supervision, Funding acquisition, Validation, Writing – original draft, Project administration, Writing – review and editing

### Author ORCIDs

Zigmunds Orlovskis (ID) https://orcid.org/0000-0001-8583-9814
Archana Singh (ID) https://orcid.org/0000-0002-5027-4582
Adi Kliot (ID) https://orcid.org/0000-0002-0370-0817
Saskia A Hogenhout (ID) https://orcid.org/0000-0003-1371-5606

Reviewer #1 (Public review): https://doi.org/10.7554/eLife.98992.3.sa1
Reviewer #2 (Public review): https://doi.org/10.7554/eLife.98992.3.sa2
Author response https://doi.org/10.7554/eLife.98992.3.sa3

## Additional files

**Supplementary files**

Supplementary file 1. FPKM and differential expression values of 17'153 genes included in the response analyses of plants to SAP54 vs GFP and male vs female leafhopper exposure.

Supplementary file 2. IDs and log2-fold changes of differentially expressed genes (DEGs) of male and female *M. quadrilineatus* leafhopper-exposed GFP and SAP54 plants compared to insect free GFP plants. Tab A: GFP plants exposed to caged females versus cages alone. Tab B: GFP plants exposed to caged males versus cages alone. SAP54 plants exposed to caged females versus cages alone. Tab D: SAP54 plants exposed to caged males versus cages alone. Tab E: SAP54 versus GFP plants exposed to cages alone.

Supplementary file 3. MapMan build-in functional bins enriched for DEGs in male and female *M. quadrilineatus* leafhopper-exposed GFP and SAP54 plants compared to insect-free GFP plants.

Supplementary file 4. MapMan build-in biotic stress functional bins enriched for DEGs in male and female *M. quadrilineatus* leafhopper-exposed GFP and SAP54 plants compared to insect-free/cage-only GFP plants. Tab A: MapMan build-in biotic stress functional bins enriched for DEGs. *p*-values based on Wilcoxon rank test with BH correction. Bins highlighted in bold are enriched for DEGs. Individual transcript identities within significantly enriched bins are provided in the tabs B, C, D, E. Tab B: list of DEGs within significantly enriched biotic stress bins for GFP plants exposed to caged females versus cages alone. list of DEGs within significantly enriched biotic stress bins for GFP plants exposed to caged males versus cages alone. Tab D: list of DEGs within significantly enriched biotic stress bins for SAP54 plants exposed to caged females versus cages alone. Tab E: list of DEGs within significantly enriched biotic stress bins for SAP54 plants exposed to caged males versus cages alone.

Supplementary file 5. Manually curated and assigned defence signalling bins for MapMan import. Tab A, Functional bin categories. Tab B, IDs of genes assigned to each of the functional bins listed in Tab A. ''Type T'' means ''transcript'', nomenclature used according to in-built MapMan pathway files.

Supplementary file 6. Functional bins for manually annotated defence genes enriched for DEGs in male and female *M. quadrilineatus* leafhopper-exposed GFP and SAP54 plants compared to insect free GFP plants. Tab A: Functional bins for manually annotated defense genes enriched for DEGs. P-values based on Wilcoxon rank test with BH correction. Bins highlighted in bold are enriched for DEGs. Individual transcript identities within significantly enriched bins are provided in the tabs B, C, D, E. Tab B: list of DEGs within significantly enriched biotic stress bins for GFP plants exposed to caged females versus cages alone. list of DEGs within significantly enriched biotic stress bins for GFP plants exposed to caged males versus cages alone. Tab D: list of DEGs within significantly enriched biotic stress bins for SAP54 plants exposed to caged females versus cages alone. Tab E: list of DEGs within significantly enriched biotic stress bins for SAP54 plants exposed to caged males versus cages alone.

Supplementary file 7. MapMan build-in functional bins enriched for DEGs in SAP54 versus GFP plants with or without exposure to male and female *M. quadrilineatus* leafhoppers.

Supplementary file 8. Biotic stress bins from MapMan build-in and manually curated defence signalling pathway bins enriched for DEGs in GFP and SAP54 plants with or without exposure to male and female *M. quadrilineatus* leafhoppers.

Supplementary file 9. Fold-expression changes of MADS-box transcription factor genes insect-exposed SAP54 vs GFP leaves. Tab A: List of 20 genes encoding MADS-box transcription factors that are expressed in leaves in the (insect-exposed) SAP54 and GFP plants. Tab B: List of all 107 genes annotated as MADS-box transcription factors in *A. thaliana* (**de Folter et al., 2005**).

Supplementary file 10. DEG encoded function enrichment for all MapMan built-in bins in female- or male-exposed *svp* plants vs female- or male-exposed wild type plants. *p*-values based on Wilcoxon rank test and have BH correction.

Supplementary file 11. DEG encoded function enrichment for MapMan built-in built-in biotic stress and manually designed defence signalling pathways in female- or male-exposed *svp* plants

vs female- or male-exposed wild type plants. Tab A: biotic stress and defence signalling pathway bins enriched with female- or male-specific DEGs. *p*-value based on Wilcoxon rank test and BH correction. Tab B: list if DEGs within enriched bins for clip-caged females on *svp* plants vs clip-caged females on wild type plants. list if DEGs within enriched bins for clip-caged males on *svp* plants vs clip-caged males on wild type plants. Tab D: since no enriched bins found for male-specific DEGs in tab A, no transcripts highlighted here.

Supplementary file 12. GO-term and MapMan functional enrichment on shared 155 DEGs between ''male-exposed SAP54 vs male-exposed GFP'' comparison and ''male-exposed *svp* vs male-exposed wild type'' comparison. Tab A: GO-term enrichment with 155 DEGs shared by male-exposed SAP54 plants (versus male-exposed GFP plants) and male-exposed *svp* plants (vs male-exposed wild type plants) and depicted in *Figure 6—figure supplement 2D*. Tab B: gene names, encoded function description and log2(fold) change for the 155 DEGs analysed in tab A and depicted in *Figure 6— figure supplement 2D*. Common functions identified between (1) MapMan function enrichment for 3816 DEGs on male-exposed SAP54 plants vs male-exposed GFP plants and (2) MapMan function enrichment for 807 DEGs on male-exposed *svp* plants vs male-exposed wild type plants; Venn diagram of the DEGs in *Figure 6—figure supplement 2D*.

MDAR checklist

## Data availability

RNA-seq raw data (fastq) are uploaded to NCBI under BioProject: PRJNA1090849.

The following dataset was generated:

| Author(s) | Year | Dataset title | Dataset URL | Database and Identifier |
|---|---|---|---|---|
| Orlovskis Z, Singh A, Kliot A, Huang W, Hogenhout SA | 2024 | *Arabidopsis thaliana* (thale cress) | https://www.ncbi.nlm.nih.gov/bioproject?term=PRJNA1090849 | NCBI BioProject, PRJNA1090849 |

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
