## [Editor Report · eLife Assessment]

This study highlights an **important** discovery: a bacterial pathogen's effector influences plant responses that in turn affect how the leafhopper insect vector for the bacteria is attracted to the plants in a sex-dependent manner. The research is backed by **convincing** physiological and transcriptome analyses. This study unveils a complex interdependence between the pathogen effector, male leafhoppers, and a plant transcription factor in modulating female attraction to the plant, shedding light on previously unexplored aspects of plant-bacteria-insect interactions.

---

## [Referee Report · Reviewer #1 (Public review)]

Summary:

Orlovski and his colleagues revealed an interesting phenomenon that SAP54-overexpressing leaf exposure to leafhopper males is required for the attraction of followed females. By transcriptomic analysis, they demonstrated that SAP54 effectively suppresses biotic stress response pathways in leaves exposed to the males. Furthermore, they clarified how SAP54, by targeting SVP, heightens leaf vulnerability to leafhopper males, thus facilitating female attraction and subsequent plant colonization by the insects.

Strengths:

The phenomenon of this study is interesting and exciting.

---

## [Referee Report · Reviewer #2 (Public review)]

Summary:

In this study, the authors show that leaf exposure to leafhopper males is required for female attraction in the SAP54-expressing plant. They clarify how SAP54, by degrading SVP, suppresses biotic stress response pathways in leaves exposed to the males, thus facilitating female attraction and plant colonization.

Strengths:

This study suggests the possibility that the attraction of insect vectors to leaves is the major function of SAP54, and the induction of the leaf-like flowers may be a side-effect of the degradation of MTFs and SVP. It is a very surprising discovery that only male insect vectors can effectively suppress the plant's biotic stress response pathway. Although there has been interest in the phyllody symptoms induced by SAP54, the purpose and advantage of secreting SAP54 were unknown. The results of this study shed light on the significance of secreted proteins in the phytoplasma life cycle and should be highly evaluated.

Weaknesses:

There are no major weaknesses. The mechanism behind why only male leafhoppers reduce plant defense responses in the presence of SAP54 remains somewhat unclear, but clarifying this is beyond the scope of this study and is for future work.

---

## [Author Response]

The following is the authors’ response to the original reviews.

**Public Reviews:**

**Reviewer #1 (Public Review):**
Summary:Orlovskis and his colleagues revealed an interesting phenomenon that SAP54-overexpressing leaf exposure to leafhopper males is required for the attraction of followed females. By transcriptomic analysis, they demonstrated that SAP54 effectively suppresses biotic stress response pathways in leaves exposed to the males. Furthermore, they clarified how SAP54, by targeting SVP, heightens leaf vulnerability to leafhopper males, thus facilitating female attraction and subsequent plant colonization by the insects.Strengths:The phenomenon of this study is interesting and exciting.Weaknesses:The underlying mechanisms of this phenomenon are not convincing.

We thank the reviewer for the comment of finding our study interesting and exciting. However, we respectfully disagree with the reviewer assertion that the mechanisms we uncovered are unconvincing.

We have uncovered a significant portion of the mechanisms by which SAP54 induces the leafhopper attraction phenotype.

First, we discovered that the SAP54-mediated attraction of leafhoppers requires the presence of male leafhoppers on the leaves. Female leafhoppers were only attracted and laid more eggs on leaves when both SAP54 and male leafhoppers were present. In the absence of either males or SAP54, female leafhoppers did not exhibit this behaviour.

Second, we found that biotic stress responses in leaves were significantly downregulated when exposed to SAP54 and male leafhoppers, with a much lesser effect observed in the presence of females.

Third, we identified that the presence of the MADS-box transcription factor SHORT VEGETATIVE PHASE (SVP) in leaves is crucial for the leafhopper attraction phenotype, and that SAP54 facilitates the degradation of SVP.

Our research corroborates previous findings that SAP54-mediated degradation of MADS-box transcription factors depends on the 26S proteasome shuttle factor RAD23, which we found previously to also be necessary for the leafhopper attraction phenotype (MacLean et al., 2014. PMID: 24714165). This finding has been replicated by other research groups. Previous research has also revealed that leafhoppers are specifically attracted to leaves, not to the leaf-like flowers (Orlovskis & Hogenhout, 2016. PMID: 27446117).

Collectively, these results suggest that SAP54 acts as a "matchmaker", helping male leafhoppers locate mates more easily by degrading SVP-containing complexes in leaves. We have updated the model in Fig. 7 to better illustrate our findings.

**Reviewer #2 (Public Review):**
Summary:In this study, the authors show that leaf exposure to leafhopper males is required for female attraction in the SAP54-expressing plant. They clarify how SAP54, by degrading SVP, suppresses biotic stress response pathways in leaves exposed to the males, thus facilitating female attraction and plant colonization.Strengths:This study suggests the possibility that the attraction of insect vectors to leaves is the major function of SAP54, and the induction of the leaf-like flowers may be a side-effect of the degradation of MTFs and SVP. It is a very surprising discovery that only male insect vectors can effectively suppress the plant's biotic stress response pathway. Although there has been interest in the phyllody symptoms induced by SAP54, the purpose, and advantage of secreting SAP54 were unknown. The results of this study shed light on the significance of secreted proteins in the phytoplasma life cycle and should be highly evaluated.Weaknesses:One weakness of this study is that the mechanisms by which male and female leafhoppers differentially affect plant defense responses remain unclear, although I understand that this is a future study.The authors show that female feeding suppresses female colonization on SAP54-expressing plants. This is also an intriguing phenomenon but this study doesn't explain its molecular mechanism (Figure 7).

Strengths:

We appreciate the reviewer's assessment of the strengths of our study. We do indeed discuss the possibility that the induction of leaf-like flowers could be a side effect of the SAP54 effector function. However, it is not uncommon for effectors to have multiple functions, as has been frequently demonstrated for viral proteins (e.g., PMID: 34618877). Furthermore, it is increasingly evident that developmental and immune processes in organisms often overlap and are mediated by the same proteins. A notable example is the Toll-like receptors, which are widely recognized for their role in innate immunity but were initially discovered for their involvement in various developmental processes (e.g., PMID: 29695493).

MADS-box transcription factors are known to regulate various developmental pathways in plants, and their diversification has been a key driver of evolutionary innovations in plant development. These factors are comparable to HOX genes, which are essential for the development of bilateral animals. While the role of MADS-box transcription factors in orchestrating flowering has been well-documented, recent evidence has emerged showing that they also play a role in regulating immune processes in plants. Our findings contribute to this emerging understanding, presenting novel insights into the multifunctional roles of these transcription factors.

Specifically, the MADS-box transcription factor SVP has vital roles in both plant immunity and flowering. The SAP54-mediated targeting of this transcription factor may therefore confer multiple advantages to phytoplasmas that, as obligate colonisers, depend on plants and transmission by insects for survival. Firstly, the inhibition of flowering could delay plant senescence and death, which is particularly relevant in annual plants, the primary hosts of AY-WB phytoplasma studied here. Secondly, the downregulation of plant defence responses, particularly against males, facilitates the attraction of females, which are more likely to reproduce and thus increase the number of vectors for phytoplasma transmission. Given that phytoplasmas are obligate organisms with highly reduced genomes, it is plausible that they rely on ‘efficient proteins’ capable of targeting multiple key pathways in their hosts.

Weaknesses:

As explained above, we have uncovered a substantial portion of the mechanisms through which SAP54 induces the leafhopper attraction phenotypes that includes the identification of MADS-box transcription factor SVP as an important contributor. We have updated the model in Fig. 7 to better illustrate our findings.

It is known that SVP forms quaternary structures with other (MADS-box) transcription factors, and it is seems likely that the degradations of specific SVP complexes present in fully developed leaves play a significant role in the downregulation of immune genes in the presence of SAP54 and males. These specific complexes also do not form in *svp* mutants, which could explain why females are attracted to these mutant plants in the presence of males. However, transcription profiles are different in male-exposed SAP54 vs male-exposed *svp* plants. This may be explained by SVP having multiple functions, including those that are not targeted by SAP54.

Identifying which SVP complexes contribute to the male-mediated downregulation of immunity in the presence of SAP54 would require the development of a broad range of tools to investigate plant immunity without the confounding effects of developmental changes. This line of inquiry extends beyond the findings presented in this study.

**Recommendations for the authors:**

**Reviewer #1 (Recommendations For The Authors):**
Orlovskis and colleagues revealed an interesting phenomenon that SAP54-overexpressing leaf exposure to leafhopper males is required for the attraction of followed females. By transcriptomic analysis, they demonstrated that SAP54 effectively suppresses biotic stress response pathways in leaves exposed to the males. Furthermore, they clarified how SAP54, by targeting SVP, heightens leaf vulnerability to leafhopper males, thus facilitating female attraction and subsequent plant colonization by the insects. The discovery of this study is interesting and exciting. However, I have a few concerns that require authors to address.(1) The author demonstrated that SAP54-overexpressing leaf exposure to leafhopper males is more attractive to females. However, I was confused that the author did not analyse the choice preference of males. This is important, as the author demonstrated later that "SAP54 plants exposed to males display significant downregulation of biotic stress responses". It is very possible that the female is attracted by a mating signal, but not by reduced biotic stress responses. Also, it is important to address whether the female used in this study is virgin.

We have analysed male preference in feeding choice tests (Figure 1, treatment 3) and described our findings in the text (p7; lines 214-216). For added clarity, we have revised the text on p7 (lines 214-216) to specify that males alone do not show any feeding preference for SAP54 plants.

Additionally, we investigated whether females could be attracted to male-exposed SAP54 plants prior to landing and feeding using choice experiments, as depicted in Supplemental Figure 3 and discussed in the text (p9; lines 265-271). These findings suggest that long-distance cues alone do not fully account for the female attraction phenotype observed in Figure 1. We acknowledge that mating calls or volatiles may complement or enhance the transcriptional changes in male-exposed SAP54 leaves. This interpretation is further supported by comparing Figure 1, treatments 4 and 5, which shows that removing males from SAP54 leaves before female choice does not increase female colonisation. To enhance clarity and precision, we have added the term "solely" to the results (p9; line 265) and discussion (p25; line 719), and included a new sentence on p26 (lines 726-730): "However, given that the removal of males from SAP54 leaves prior to female choice does not enhance female colonisation (comparison of Figure 1, treatment 4 with treatment 5), we cannot exclude the possibility that male-produced volatiles or mating calls could enhance or supplement SAP54-dependent changes in biotic stress responses to males, thereby enhancing female attraction."

We have also updated the methods section to clarify that a mixture of virgin and pre-mated females was used in all experiments (p28; lines 798-799), consistent with our previously published work (Orlovskis & Hogenhout, 2016. PMID: 27446117; MacLean et al., 2014. PMID: 24714165).

(2) I was confused by the rationality of the section "Female leafhopper preference for male-exposed SAP54 plants unlikely involves long-distance cues". The volatile cues or mating calls from males can be only perceived from a distance?

As mentioned in our response to comment 1, for clarity, we have added new text to both the results (p9; line 265) and discussion sections (p25; lines 719 and 726-730). In the results section highlighted by the reviewer (p8-9), we aimed to explicitly test whether cues produced by males (such as mating calls or pheromones) or SAP54 plants (such as plant volatiles) could account for female attraction from a distance, independent of, and prior to, physical contact with the plants or male insects.

To address the possibility that volatiles or mating calls might be perceived simultaneously with downregulated biotic stress responses, we have included an additional sentence in the discussion, which addresses comments 1 and 2 from the reviewers. Furthermore, it is important to note that Figure 1, treatment 4, mirrors the results of Figure 1, treatment 1, suggesting that direct physical contact between males and females is not necessary for the observed female attraction. This conclusion, derived from our experiments, was already emphasised in the main text (p7; lines 218-222).

(3) Line 271-273. How the author concluded the "immediate access". A time course experiment (detect the number of insects on each plant at different time point) for host-choice experiment is necessary.

We have corrected and rephrased the sentence as follows:

‘’Therefore, these results indicate that female reproductive preference for the male-exposed SAP54 versus GFP plants is dependent on immediate access of the direct females access to the leaves of SAP54 plants and presence of males on these leaves.’’ (p9; lines 267-271).

(4) I appreciate the transcriptome analysis. However, the figures are poorly organized. i.e. the heatmap in Figure 2 was poorly understood. The author should clearly address what is upregulated or downregulated. It is meaningless to exhibit the heatmap without explaining what gene represented. Also, it is hard for readers to distinguish the difference between the 4 maps in Figure 2, similar to the two figures in Figure 3.

We thank the reviewer for the recommendation. To make Figure 2 and 3 easier to read and understand as stand-alone, we have changed and improved the corresponding figure legends, highlighting the colouring of up- and down-regulated DEGs as well as explaining the related supplementary file content in figure legends. For brevity and clarity, we have removed the mentioning of figure supplement 4, 5 and 6 as they have already been explained and referred to in the main text but do not directly relate to Figure 2 or 3 but rather data processing prior to analysis in Figure 2.

We hope that the improvements in figure legends will make the Figures 2 and 3 easier and quicker to understand.

(5) For transcriptomic analysis, three out of four replicates were well clustered, and the author excluded the outliers in subsequent analysis. Is this treatment commonly used in transcriptomic analysis? If yes, please provide corresponding references.

Removing outliers from transcriptomic data is not unusual, as it enhances the classification of treatment groups and increases the efficiency of detecting biologically relevant differentially expressed genes (DEGs) (PMID: 36833313; PMID: 32600248). For large datasets, especially in clinical studies, automated procedures and algorithms have been developed for this purpose (PMID: 32600248; doi.org/10.1101/144519). Given our relatively small sample size of 4, we opted for a PCA-based manual outlier evaluation, followed by repeated PCA without the identified outliers. This approach demonstrated improved group discrimination (Figure Supplement 4), which can enhance downstream characterization of DEGs and pathways that explain female preference for male-exposed SAP54 plants. We have detailed this procedure on pages 9-10. It is worth noting that other automated outlier removal methods, which are also based on PCA, have been shown to be as effective as manual outlier removal (PMID: 32600248).

(6) Figure 5A. How the experiment was done? The HA-SVP and other HA-tagged genes were stably or transiently expressed in GFP and GFP-SAP54 plants? How many replicates were conducted? The band intensity from different biological replicates should be provided. In this manuscript, no information is provided even in the method section.

We thank the reviewer for noticing this and have updated the methods section providing more details on transient protoplast expression assays (p39; line 835). We have performed two independent degradation assays for all 5 MTF proteins and indicated in the legend of Figure 5. Western blot results from both experiments are provided as a new figure supplement 10 (p53). The degradation/destabilisation efficiency was calculated as the HA intensity divided by the RuBisCo large subunit (rbcL) intensity from the same sample, normalised to the intensity of the sample with the highest ratio from the same leaf (Rel HA/rbcL) using ImageJ. Relative pixel intensities are provided above each treatment in new figure supplement 10, as requested by the reviewer.

(7) For the interaction assay, only Y2H was conducted. Generally, at least two methods are needed to confirm protein interaction. This is also applicable to degradation assays.

There is substantial prior evidence that SAP54 interacts with MADS-box transcription factors and facilitates their degradation in plants, a process that also involves the 26S proteasome shuttle factor RAD23 (MacLean et al., 2014; PMID: 24714165). This interaction has been independently confirmed by other research groups using various methods, including split-YFP assays (e.g., PMID: 24597566, PMID: 26179462). Given the extensive data already available on this topic, it would be redundant to replicate all of these findings in our manuscript. Instead, we have focused on a few validated assays that effectively demonstrate the specific interactions between SAP54 and MADS-box transcription factors.

(8) Lines 528-530. No direct evidence in this study was provided for how SAP54-mediated degradation of SVP. The author should tone down the claim.

Our findings demonstrate that SVP is degraded in plant cells in the presence of SAP54. Additionally, through yeast two-hybrid assays, we show that SAP54 does not directly bind to SVP but does directly interact with several MADS-box transcription factors known to associate with SVP. We also provide evidence that they interact with SVP herein. Furthermore, previous studies have shown that SAP54 facilitates the degradation of MADS-box transcription factor complexes of *Arabidopsis* and several other eudicot species (PMID: 24597566, PMID: 26179462, PMID: 28505304, PMID: 35234248; PMID: 38105442). We have described observations herein and of others (see main text pages 4-5, pages 19-20), and believe that we have presented them accurately without overstating our conclusions.

(9) Overall, the phenomenon of this study is interesting, but the underlying mechanisms are not solidified. Additional work is still needed in future studies.

We respectfully disagree—we have identified a significant portion of the mechanisms by which SAP54 induces these phenotypes. As with any research, new data often leads to further questions that may be addressed by follow-up studies. Please refer to our previous responses for additional context.

**Reviewer #2 (Recommendations For The Authors):**
Major commentIt will be interesting to see how long male feeding affects changes in gene expression in plants. No feeding choice of females was observed on the SAP54 plants when males were removed from the clip-cages prior to the choice test with females alone (Figure 1, Treatment 5; Figure Supplement 1, Treatment 5). This indicates that SAP54 plants lose their ability to attract females as soon as males are removed. On the other hand, if the suppression of the plant's stress response pathway by male feeding continues for some time even after males are removed, I think that we cannot exclude the possiblity that volatiles emitted by males may partially promote female feeding and colonization.

As described above, our findings suggest that long-distance cues alone do not fully account for the female attraction phenotype observed in Figure 1. We acknowledge that mating calls or volatiles may complement or enhance the transcriptional changes in male-exposed SAP54 leaves. This interpretation is further supported by comparing Figure 1, treatments 4 and 5, which shows that removing males from SAP54 leaves before female choice does not increase female colonisation. To enhance clarity and precision, we have added the term "solely" to the results (p9; line 265) and discussion (p25; line 719), and included a new sentence on p26 (lines 726-730): "However, given that the removal of males from SAP54 leaves prior to female choice does not enhance female colonisation (comparison of Figure 1, treatment 4 with treatment 5), we cannot exclude the possibility that male-produced volatiles or mating calls could enhance or supplement SAP54-dependent changes in biotic stress responses to males, thereby enhancing female attraction."

Minor commentsThe legend of Figure 1 is missing an explanation for panel C.

Thank you for noticing this. We have added the missing information.

Although from a different perspective from this study, a relationship between phytoplasma infection and SVP has been previously reported (Yang et al., Plant Physiology, 2015). Shouldn't this paper be cited somewhere?

We thank the reviewer for identifying this oversight. We have added the missing reference (PMID: 26103992) and clarified that, as seen in Figure 5E (p20; lines 555-558), our findings show a similar upregulation of SVP in male-exposed SAP54 plants as reported by Yang et al. This suggests that SAP54 and its homologs, such as PHYL1, may indeed operate through similar mechanisms by targeting MTFs that are crucial for their function. While Yang et al. described the role of SVP in the development of abnormal flower phenotypes in Catharanthus, our study reveals a completely novel role for SVP in plant-insect interactions. Although SAP54 destabilises the SVP protein, its transcript is upregulated in the presence of SAP54, indicating a potential disruption of MTF autoregulation and the MTF network as a whole.